**EMBO** *reports*

# Inflammation and IL-4 regulate Parkinson's and Crohn's disease associated kinase LRRK2

Dina Dikovskaya [1,2✉], Rebecca Pemberton[1], Matthew Taylor [1,5], Anna Tasegian[1,6], Purbasha Bhattacharya [1], Karolina Zeneviciute [1], Esther M Sammler [1,3], Andrew J M Howden [4], Dario R Alessi [1] & Mahima Swamy [1✉]

## Abstract

Mutations in Leucine-Rich Repeat protein Kinase 2 (LRRK2) are associated with Parkinson's disease (PD) and Crohn's disease (CD), but the regulation of LRRK2 during inflammation remains relatively unexplored. Here we describe the development of a flow cytometry-based assay to assess LRRK2 activity in individual cells and the generation of an *EGFP-Lrrk2* knock-in reporter mouse to analyse cell-specific LRRK2 expression. Using these tools, we measured LRRK2 levels and activity in murine splenic and intestinal immune cells and in human blood. Anti-CD3 induced inflammation increases LRRK2 expression and activity in B cells and monocytes, while in mature neutrophils, inflammation stimulates activity but reduces LRRK2 expression. A kinase-activating PD-associated LRRK2-R1441C mutation exacerbates inflammation-induced activation of LRRK2 specifically in monocytes and macrophages. We identify IL-4 as a novel T-cell-derived factor that upregulates LRRK2 expression and activity in B cells, replicating inflammatory effects observed in vivo. Our findings provide valuable new insights into the regulation of the LRRK2 pathway in immune cells, crucial for understanding LRRK2 and its therapeutic potential in inflammatory diseases such as CD.

**Keywords** LRRK2; Crohn's Disease; Parkinson's Disease; B Cells; Interleukin-4
**Subject Categories** Immunology; Molecular Biology of Disease; Signal Transduction

## Introduction

Leucine-Rich Repeat protein Kinase 2 (LRRK2) is a large multi-domain protein that combines kinase and GTPase enzymatic activities, involved in regulation of multiple intracellular processes such as lysosomal maintenance and functioning, vesicular trafficking (Kuwahara and Iwatsubo, 2020; Roosen and Cookson, 2016), mitophagy (Singh et al, 2021), inflammasome activation (Liu et al, 2017a) as well as several other innate immune pathways (Ahmadi Rastegar and Dzamko, 2020). Many LRRK2 functions are thought to be mediated by its kinase substrates Rab GTPases (Rab1, Rab3, Rab5, Rab8, Rab10, Rab12, Rab29, Rab35 and Rab43) (Alessi and Pfeffer, 2024). Mutations in LRRK2 are causal for Parkinson's disease (PD) (Paisan-Ruiz et al, 2004; Zimprich et al, 2004) and genetically associated with several inflammatory diseases including Crohn's disease (CD) (Hui et al, 2018; Witoelar et al, 2017), leprosy (Fava et al, 2016; Wang et al, 2015) and systemic lupus erythematosus (SLE) (Zhang et al, 2017). Most of the well-characterised disease-associated mutations in LRRK2 cluster within either its kinase domain or a ROC-COR domain and stimulate kinase activity (Kalogeropulou et al, 2022), making LRRK2 kinase an attractive drug target. In animal testing studies including non-human primates, high doses of LRRK2 kinase inhibitors that ablate pathway activity revealed lung and kidney abnormalities that are likely due to on-target effects as these are recapitulated in the LRRK2 knock-out (KO) phenotype (Araki et al, 2018; Wojewska and Kortholt, 2021). However, a recent Phase-1 clinical study with an inhibitor (DNL201) that suppressed LRRK2 pathway activity by ~70%, concluded that this compound was safe and well tolerated, warranting further clinical development of LRRK2 inhibitors as a therapeutic modality for PD (Jennings et al, 2022).

However, the potential effect of LRRK2 inhibition on specific cell types and their functions in the context of inflammatory disease is not well understood. LRRK2 is highly expressed in select human immune cells, including neutrophils, monocytes and B cells, and its level has been reported to be further increased in B cells, T cells and monocytes after prolonged IFNγ stimulation (Ahmadi Rastegar et al, 2022; Cook et al, 2017; Gardet et al, 2010; Hakimi et al, 2011; Kuss et al, 2014; Thevenet et al, 2011). While the role of LRRK2 and the consequences of LRRK2 inhibition have been extensively studied in monocytes and macrophages (Dzamko et al, 2015; Herbst et al, 2020; Lee et al, 2020; Thevenet et al, 2011; Yadavalli and Ferguson, 2023), little is known about other cells with active LRRK2. Furthermore, the expression and functions of LRRK2 in intestinal immune cells, relevant to CD, are unknown. This knowledge is critical to understand how LRRK2

[1]Medical Research Council (MRC) Protein Phosphorylation and Ubiquitylation Unit, School of Life Sciences, University of Dundee, Dow Street, Dundee DD1 5EH, UK. [2]Peninsula Medical School, University of Plymouth, Drake Circus, Plymouth PL4 8AA, UK. [3]Molecular and Clinical Medicine, Ninewells Hospital and Medical School, University of Dundee, Dundee DD1 9SY, UK. [4]Cell Signalling and Immunology, School of Life Sciences, University of Dundee, Dow Street, Dundee DD1 5EH, UK. [5]Present address: GlaxoSmithKline, Stevenage, UK. [6]Present address: Amphista Therapeutics Ltd., Granta Park, Great Abington, Cambridge CB21 6GQ, UK. ✉E-mail: Dina.dikovskaya@plymouth.ac.uk; m.swamy@dundee.ac.uk

inhibitors impact immune cells and their ability to trigger and mediate inflammatory responses.

Here, to establish the landscape of LRRK2 activity among different immune cell types, we developed and validated a flow cytometry-based assay that measures LRRK2-dependent phosphorylation of its substrate Rab10 with single-cell resolution. We combined it with a newly developed *EGFP-Lrrk2* reporter mouse to better quantify the relative expression levels of LRRK2 protein in mouse splenic and intestinal immune cells and determine how these are affected by the intestinal tissue environment and inflammation. We demonstrate that the PD-associated R1441C mutation in LRRK2 not only enhances the basal LRRK2 activity in most tested LRRK2-expressing immune cells but also exacerbates eosinophils, immature monocytes and macrophages response to inflammation. We further demonstrate that in B cells, inflammation-driven increases in level and activity of LRRK2 can be recapitulated in vitro, accompanied by a threefold increase in *Lrrk2* mRNA. We identify Interleukin-4 (IL-4) as the cytokine responsible for inducing and activating LRRK2 in B cells.

# Results

## Development of a flow cytometric assay for LRRK2 activity in tissues

To identify cells that express LRRK2 in intestinal tissues, the primary site affected by CD, we first attempted to use immuno-fluorescence (IF) in frozen or paraffin-embedded intestinal preparations from wild-type (WT) mice, controlled by analogous staining in mice deficient for LRRK2 (LRRK2-KO) to establish the specificity of staining. Despite previous reports of successful detection of LRRK2 in mouse brain and gut tissues by IF and immunohistochemistry (IHC) (Davies et al, 2013; West et al, 2014; Zhang et al, 2015), we were unable to detect LRRK2-specific staining in mouse tissues with several published anti-LRRK2 antibodies, including c41-2, the most commonly used antibody for LRRK2 in IF. Indeed, the c41-2-elicited signal within the ileal lamina propria (Fig. EV1A, left panels), in Peyer's patches rich in immune cells (Fig. EV1A, second panel) or in the lung (Fig. EV1B) was not diminished in tissues obtained from LRRK2-KO mice. To assess the specificity of c41-2 for LRRK2 in IF in a murine cell line, we analysed a mouse small intestinal cell line, MODE-K, that endogenously expresses high levels of LRRK2, and a LRRK2-deficient MODE-K clone made by CRISPR deletion. Immunoblotting with the highly specific LRRK2 N241A/34 antibody revealed loss of LRRK2 and loss of phosphorylation of LRRK2 kinase substrate Rab10 at residue T73 in the CRISPR LRRK2-KO clone, as expected (Fig. 1A). However, IF analysis of LRRK2 using c41-2 showed no difference in LRRK2 staining in PFA- or methanol-fixed LRRK2-WT and LRRK2-deficient MODE-K cells (Fig. EV2A). Among several tested LRRK2 antibodies, N241A/34 displayed the highest specificity for LRRK2 detection in MODE-K cells, particularly when combined with methanol fixation (Fig. EV2B). However, IF staining of mouse ileum with the N241/34 antibody was still unspecific, since it generated equally strong signals in LRRK2-KO tissues (Fig. EV3).

As an alternative approach, we explored flow cytometry-based detection. We tested several anti-LRRK2 antibodies and antibodies against phosphorylated LRRK2 substrates and selected an antibody against pThr73-Rab10, clone MJF-R21-22-5 (Lis et al, 2018), that showed the best specificity. Rab10 is the most highly phosphorylated Rab substrate of LRRK2 in most mouse tissues except the brain (Nirujogi et al, 2021). To ensure that the signal we observed was LRRK2-dependent, we included a control in which cells were treated with MLi-2, a specific LRRK2 kinase inhibitor (Fell et al, 2015). The pRab10 signal in both MLi-2 and vehicle-treated cells was above the signal generated by the secondary antibodies in the absence of primary antibodies (Fig. 1B). Importantly, in wild-type (WT) MODE-K cells, a sizeable increase in pRab10 fluorescence intensity was observed in vehicle- compared to MLi-2-treated cells. In contrast, in LRRK2-KO MODE-K cells, the pRab10 fluorescence intensity was low and unaffected by MLi-2 treatment (Fig. 1B). We defined LRRK2 activity (LRRK2-dependent pRab10) in each cell as the difference between pRab10 geometric mean fluorescence intensity (gMFI) in the presence and absence of MLi-2:

$$\text{LRRK2 activity} = \text{pRab10 gMFI}_{\text{DMSO}} - \text{pRab10 gMFI}_{\text{MLi-2}}$$

Thus, we established an assay that measures overall Rab10-directed LRRK2 activity in individual cells.

We next optimised this method to simultaneously measure LRRK2 activity in a complex mixture of cells isolated from tissues, using mouse spleen as a model organ. As a control, we used spleens obtained from LRRK2-KO mice (Fig. 1C). MLi-2/vehicle treatment and pRab10 staining were combined with staining for surface markers that identified a range of splenic cell types. Our staining revealed a distinct pattern of LRRK2 activity, with highest level in neutrophils and conventional dendritic cells (cDC), followed by that in eosinophils, B cells and macrophages, very low activity in monocytes and none in T cells (Fig. 1D). As expected, no LRRK2 activity was detected in splenocytes obtained from LRRK2-deficient mice (Fig. 1E), confirming the specificity of the assay. In contrast, splenocytes obtained from mice carrying the PD-associated VPS35-D620N mutation known to enhance LRRK2 activity (Mir et al, 2018) displayed strongly elevated LRRK2 activity in all cell types apart from T cells (Fig. 1F). These studies validated the flow cytometric assay for assessing LRRK2 activity at a single-cell level and defined the relative specific activity of LRRK2 in different immune mouse cells.

We also tested the assay in human peripheral blood mono-nuclear cells (PBMC) and isolated blood neutrophils, where we have previously shown high levels of LRRK2 activity (Fan et al, 2018). Despite the very low expression levels of LRRK2 found in human B cells compared to monocytes (Fan et al, 2018), we saw comparably low levels of LRRK2-dependent pRab10 signal in B cells and monocytes (Fig. 1G). Interestingly, very low but measurable LRRK2 activity was also seen in PBMC T cells. As expected, we saw high levels of LRRK2 activity in human neutrophils. Thus, in both human blood and murine spleens, neutrophils displayed the highest and T cells the lowest level of LRRK2 signalling.

## LRRK2 activity is regulated by tissue environment

We expanded our measurements to intestinal tissues where LRRK2 may play a role in CD. Small intestinal lamina propria (SI LP) contains similar cell types as found in the spleen. Interestingly, we found that LRRK2 activity was about 2.5-fold higher in SI LP

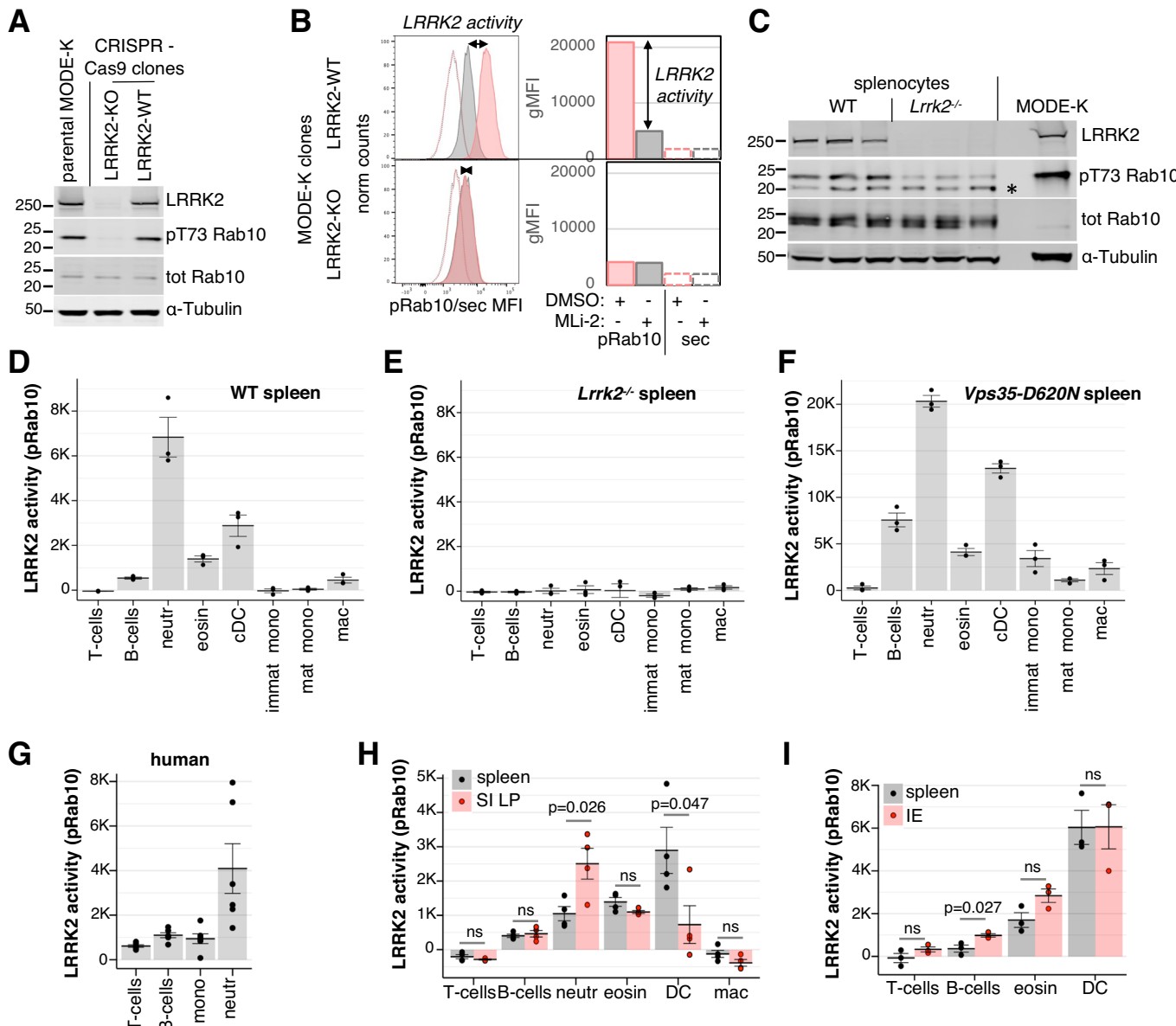

**Figure 1. LRRK2 kinase activity is differentially regulated in splenic and intestinal immune cells.**

(A) Loss of LRRK2 and phosphorylation of T73 on Rab10 in LRRK2-deficient (LRRK2-KO, second lane) CRISPR-Cas9-generated MODE-K clone. Parental MODE-K cells (left lane) and LRRK2-positive MODE-K clone (right lane) are shown as controls. Cell lysates were immunoblotted for total LRRK2, pT73 Rab10 and total Rab10, with α-Tubulin as a loading control. (B) Flow cytometry of LRRK2-WT (top) or LRRK2-KO (bottom) MODE-K clones stained with anti-pT73 Rab10 and secondary antibodies (solid lines), or with secondary antibody only (dashed lines), after 100 min incubation with 100 nM MLi-2 (grey) or vehicle (DMSO, pink). Mode-normalised histograms of fluorescence intensities are shown on the left, with geometric means of fluorescence intensities (gMFI) from the same data plotted on the right. The span of the arrows indicates LRRK2-dependent pRab10 signal, referred to henceforth as LRRK2 activity (pRab10). Data representative of three LRRK2-KO clones. (C) Isolated wild-type (WT) and LRRK2-deficient (Lrrk2⁻/⁻) littermate splenocytes immunoblotted for total LRRK2, pT73 Rab10, total Rab10 and α-Tubulin as a loading control, to confirm the loss of LRRK2 in Lrrk2⁻/⁻ cells. MODE-K cell lysate is included for comparison. A non-specific band in the pRab10 blot is indicated by *. (D–F) LRRK2 activity in WT (D) and Lrrk2⁻/⁻ (E) splenocytes (n = 3 mice each) shown in (C). Splenocytes were treated with 100 nM of MLi-2 or 0.1% DMSO and stained with surface markers for immune cell types, including T cells, B cells, neutrophils (neutr), eosinophils (eosin), conventional dendritic cells (cDC), immature monocytes (immat mono), mature monocytes (mat mono) and macrophages (mac), before staining for intracellular pRab10. See Appendix Fig. S1A for gating strategy. (D) is representative of multiple experiments. (F) LRRK2 activity in splenocytes from mice (n = 3) carrying VPS35-D620N mutation. (G) Human peripheral blood mononuclear cells and neutrophils were isolated and stained for LRRK2-dependent pRab10 (n = 6 donors) (H). LRRK2 activity in indicated cell types isolated from spleen (grey) or small intestine lamina propria (SI LP, red) from WT mice (n = 4). See Appendix Fig. S1B for gating. (I) LRRK2 activity in indicated types of splenocytes (grey) or leukocytes of the small intestinal epithelium (IE, pink) from WT mice (n = 4). Data information: (D–I) dots depict LRRK2 activity (K = x1000) in biological replicates (individual mice or human donors), with means shown as bars. Error bars indicate SEM. In (I, H): P values are from two-tailed t test. ns not significant. Source data are available online for this figure.

neutrophils than in neutrophils obtained from spleen of the same mice, whereas dendritic cells from the SI LP possessed approximately four-fold lower LRRK2 activity (Fig. 1H). LRRK2 activity was higher in several types of leukocytes that reside within small intestinal epithelium (IE) than in their counterparts obtained from spleen, with B cells showing statistically significant increase in pRab10-directed LRRK2 activity (Fig. 1I). Thus, LRRK2 activity in immune cells varies depending on the tissue context.

## A novel *EGFP-Lrrk2-KI* reporter mouse reveals tissue- and cell-type-specific expression of LRRK2

Due to the challenges of assessing LRRK2 expression by IF discussed above, we developed an *EGFP-Lrrk2* knock-in (KI) mouse line, in which the endogenous LRRK2 is N-terminally tagged with EGFP (Fig. 2A). Immunoblotting of LRRK2, GFP and pRab10 levels in WT, heterozygous (HET) and homozygous (HOM) mouse *EGFP-Lrrk2-KI* brain, spleen, kidney and large intestine lysates confirmed the presence of higher molecular weight LRRK2 protein that co-migrated with GFP signal in HET and HOM (Fig. EV4A,B; Appendix Fig. S2). We noted that the addition of the EGFP tag reduced total LRRK2 levels ~50% in brain, ~25% in spleen, without markedly affecting expression in intestine or kidney (Fig. EV4; Appendix Fig. S2). We also observed that pRab10 and pRab12 phosphorylation were reduced between ~20% to ~40% in HOM spleen, intestine, brain and kidney compared to WT. As expected, an acute 2 h administration of MLi-2 to mice markedly reduced pRab10 and pRab12 phosphorylation in all WT, HET and HOM tissues studied without impacting total levels of LRRK2 (Fig. EV4A,B; Appendix Fig. S2). We also detected up to 50% reduction in pRab10 in isolated splenocytes, measured by flow cytometry as above, particularly in dendritic cells obtained from *EGFP-Lrrk2*-KI mice, compared to their WT littermates (Fig. EV4C). These results indicate that the inclusion of an EGFP tag at the N-terminus of LRRK2 moderately impacts LRRK2 activity in most tissues and cells analysed. Note that no cleaved GFP was detected in GFP pulldowns (GFP-IP) from mouse lung tissue, indicating that the GFP signal corresponds to LRRK2 expression (Fig. EV4D). Therefore, the EGFP fluorescence could serve as a readout to assess the relative levels of LRRK2 expression in different immune cells for the first time.

Among splenic immune cells, we found that the EGFP-LRRK2 expression was highest in neutrophils, followed by eosinophils, cDC and B cells, with very low expression in macrophages and monocytes. EGFP-LRRK2 signal was undetectable in T cells (Fig. 2B). The pattern of EGFP-LRRK2 levels was in some respects different from the pattern of LRRK2 activity (pRab10) measured in splenocytes of the same mice (Fig. 2C), indicating that LRRK2 activity is not determined solely by LRRK2 expression. To further validate the use of GFP flow cytometry in *EGFP-Lrrk2*-KI mouse-derived cells as a measure of LRRK2 expression, we analysed EGFP-LRRK2 expression in four major subpopulations of splenic B cells, (transitional T1 and T2, follicular (Fo) and marginal zone (MZ)), and compared it with publicly available mass spectrometry data of LRRK2 expression in the same subsets of splenic B cells (Immunological Proteome Resource, http://immpres.co.uk/, data from Salerno et al, 2023). EGFP-LRRK2 relative expression (Fig. 2D, right) in all four subsets closely resembled the LRRK2 protein level measured by mass spectrometry in the same B-cell subsets (Fig. 2D,

left), orthogonally validating the EGFP flow cytometric signal as a readout for cellular LRRK2 expression.

We next asked how EGFP-LRRK2 expression in immune cells is affected by the intestinal environment. Intestinal eosinophils expressed less EGFP-LRRK2 than splenic eosinophils (Fig. 2E, compared with Fig. 2B). There was also a clear difference in EGFP-LRRK2 level between large and small intestinal lamina propria dendritic cells, monocytes and macrophages (Fig. 2E), supporting our earlier observation that tissue environment affects LRRK2. Furthermore, when immune cells isolated from intestinal epithelium (IE) were compared to their counterparts isolated from spleen, intestinal eosinophils showed significantly lower EGFP-LRRK2 expression, while EGFP-LRRK2 expression in B cells and dendritic cells were remarkably similar between these two tissues (Fig. 2F). These data support the conclusion that the *EGFP-Lrrk2*-KI mouse model serves as a reliable reporter of cellular LRRK2 protein expression and revealed tissue-specific and cell-type-specific regulation of LRRK2 expression.

## Inflammation alters LRRK2 activity and expression

To understand how inflammation affects LRRK2 in immune cells, we isolated splenocytes 24 h after intraperitoneal injection of anti-CD3 antibodies that activate T cells to release cytokines, driving transient inflammation (Swamy et al, 2015; Xu et al, 2021; Yaguchi et al, 2004; Zhou et al, 2004). This treatment induced inflammation apparent by significant change in cellular composition in the spleen (Fig. EV5A), as exemplified by a ~35-fold increase in immature neutrophils (Deniset et al, 2017) (Fig. EV5A,B). Interestingly, anti-CD3-induced inflammation significantly increased LRRK2 activity in B cells, neutrophils and monocytes (Fig. 3A). Measuring changes in EGFP-LRRK2 expression revealed a similar but not identical picture (Fig. 3B): the LRRK2 protein level was increased in B cells, immature neutrophils and immature monocytes, while in contrast to its activity, LRRK2 protein was significantly reduced in mature neutrophils and cDC. The increase in EGFP-LRRK2 fluorescence in B cells was not due to an increase in cell size associated with inflammation, since it was much larger than the anti-CD3 ab-induced increase in autofluorescence in WT samples (Fig. EV5C). Immunoblotting confirmed increases in both LRRK2 level and kinase activity in inflamed splenocytes, as determined by immunoblotting for total LRRK2 and Rab10 phosphorylation (Fig. 3C). We also tested a different model of inflammation, Zymosan-induced peritonitis, where injection of the *Saccharomyces cerevisiae* cell wall component, Zymosan, drives a self-resolving inflammation with recruitment of neutrophils into the peritoneal cavity (Fig. EV5D). Here too, we observed a significant increase in LRRK2-dependent pRab10 in B cells both in the peritoneal lavage and the spleen (Fig. EV5E). Interestingly, in the peritoneal cavity macrophages LRRK2 activity was reduced, while in splenic macrophages LRRK2 activity was increased after zymosan injection (Fig. EV5E). These data show that LRRK2 is affected by inflammation in a cell-specific manner and emphasise that LRRK2 kinase activity and expression are decoupled in specific cell types during an inflammatory response.

## PD-associated mutation increases basal LRRK2 activity and its activation by inflammation

We next investigated how LRRK2 activity in the context of anti-CD3-induced inflammation is affected by pathogenic LRRK2 mutations. For

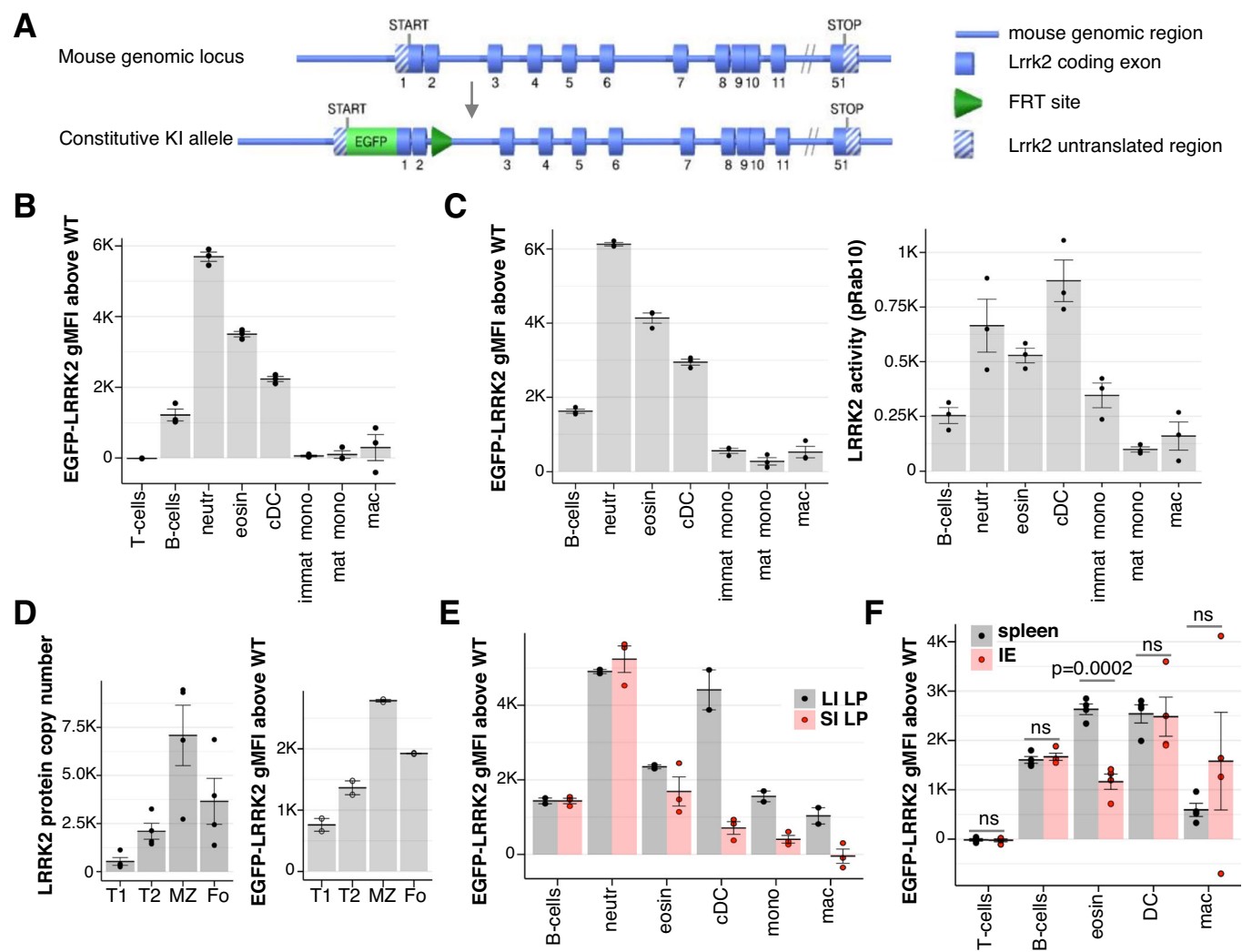

**Figure 2. Measuring LRRK2 expression using EGFP-Lrrk2-KI reporter mouse model.**

(A) Schematic of *EGFP-Lrrk2-KI* targeting strategy. *EGFP* is inserted between N-terminal UTR and the start of the ORF of *Lrrk2*, in the endogenous *Lrrk2* gene. (B) EGFP fluorescence was measured by spectral flow cytometry in homozygous *EGFP-Lrrk2-KI* ($n = 3$) splenocytes stained with cell markers and gated as in Appendix Fig. S1A, and EGFP-LRRK2 gMFI above background was calculated by subtracting the mean autofluorescence of the respective cell type measured in WT littermate mice ($n = 3$). Data representative of three separate experiments is shown. (C) EGFP-LRRK2 expression (left) and LRRK2 activity (right) was measured in *EGFP-Lrrk2-KI* mice ($n = 3$). Three WT littermates were used to determine EGFP-LRRK2 fluorescence background. (D) LRRK2 protein copy numbers (left, $n = 4$) and EGFP-LRRK2 level (right, $n = 2$) in splenic transitional 1 (T1), transitional 2 (T2), marginal zone (MZ) and follicular (Fo) B-cell subsets gated as in Appendix Fig. S3. Median LRRK2 copy numbers were quantified by label-free mass spectrometry from WT splenocytes sorted into indicated subsets (Salerno et al, 2023). Data accessed from Immunological Proteome Resource (ImmPRes; http://immpres.co.uk/). (E) EGFP-LRRK2 fluorescence in indicated lamina propria cell types isolated from large (LI LP, grey, $n = 2$) and small (SI LP, red, $n = 3$) intestine, and (F). in splenocytes (spleen, grey, $n = 4$) and cells obtained from small intestinal epithelium (IE, red, $n = 4$), measured and displayed as in (B). Data information: (B–F) Dots depict data from individual mice, with means and SEM shown as bars and error bars. Statistically significant differences calculated by one-way ANOVA are indicated with $P$ values, ns not significant. Source data are available online for this figure.

this, we exploited knock-in mice expressing the pathogenic LRRK2-R1441C mutation found in PD patients that has been previously shown to increase LRRK2 kinase activity in both murine cells and in humans (Alessi and Sammler, 2018). In both WT mice and HOM mice carrying LRRK2-R1441C mutation, anti-CD3 injection caused similar weight loss (Fig. EV5F) and increased mRNA levels of *Tnf, S100a8* and the IFN-responsive gene *Usp18* in the gut (Fig. EV5G). Moreover, cellular composition of the spleen did not show any significant changes between WT and mutant mice (Fig. EV5H), indicating that the overall level of inflammation was comparable in WT and *Lrrk2-R1441C* mice

24 h post-injection with anti-CD3. Consistent with previous data (Iannotta et al, 2020), R1441C mutation markedly increased LRRK2 kinase activity in most splenic cell types in PBS-injected mice that served as a baseline control in this experiment (Fig. 3D, compare blue and grey bars). We found that anti-CD3-mediated inflammation further increased LRRK2-R1441C activity in most cell types (except cDC and mature neutrophils), suggesting that the LRRK2-R1441C mutation and inflammation have different impacts on LRRK2 activity, either in parallel or in synergy (Fig. 3D, compare grey and red bars). Importantly, the response to anti-CD3-induced

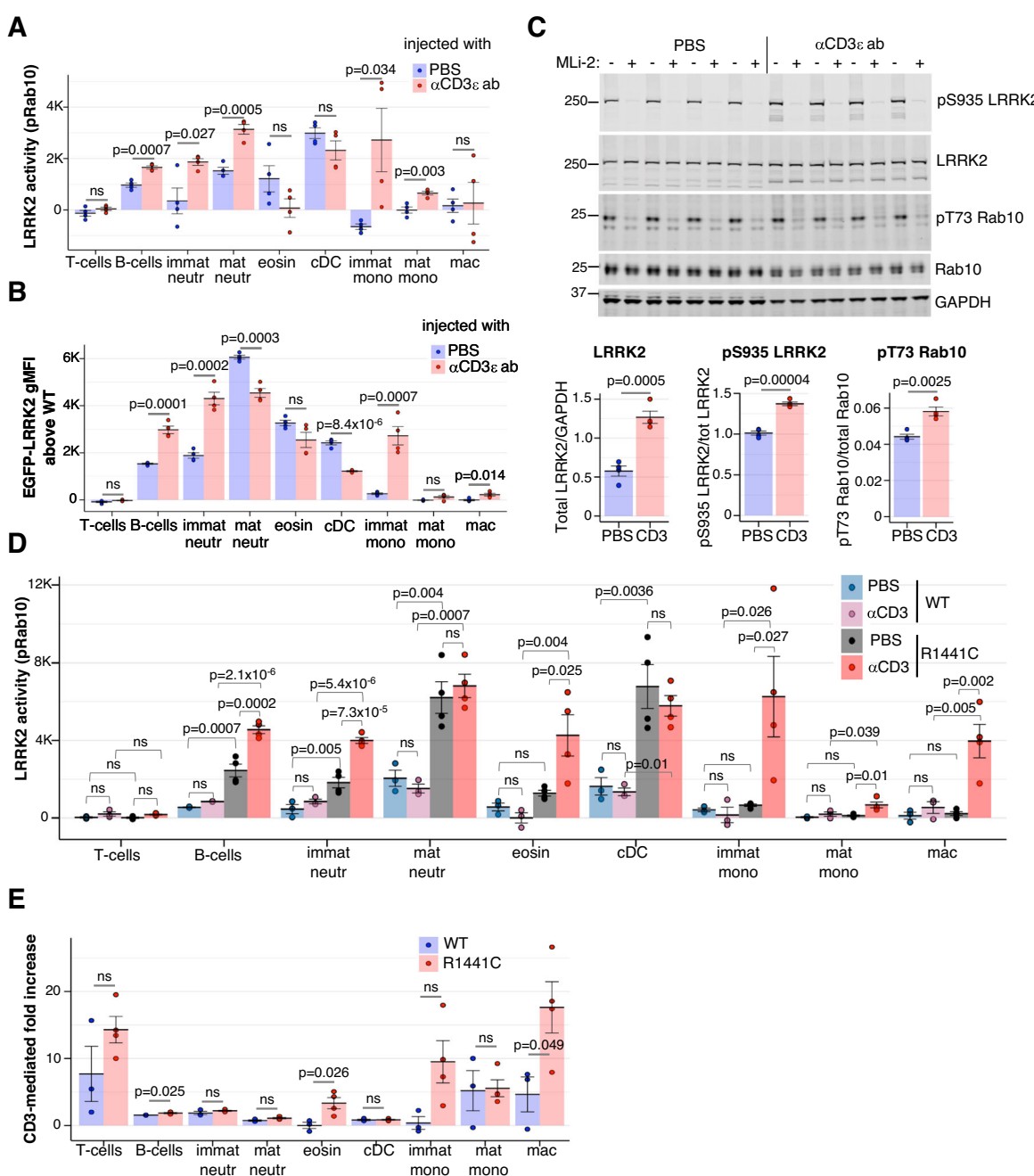

**Figure 3. LRRK2 expression level and activity is modulated in a cell-type dependent manner by inflammation.**

(A) LRRK2 activity in splenocytes isolated from mice 24 h after i.p. injection of 100 μg of anti-CD3 antibody (red, n = 4) or the same volume (100 μl) of PBS (blue, n = 4). Significant differences between PBS- and anti-CD3-injected conditions are shown by P values calculated by one-way ANOVA, ns=not significant. Representative of two independent experiments. (B) EGFP-LRRK2 fluorescence in splenocytes from *EGFP-Lrrk2-KI* mice 24 h after i.p. injection of 100 μg of anti-CD3 (red, n = 4) or PBS (blue, n = 4) co-stained with cell marker antibodies. The average autofluorescence from identically isolated, measured and analysed cells from PBS or anti-CD3-injected WT mice was subtracted as a background. Statistically significant differences are marked by P values calculated by one-way ANOVA, ns not significant. (C) Splenocytes from mice injected with either 100 μg anti-CD3 antibody or PBS shown in (A) were incubated for 30 min with media containing 100 nM MLi-2 (+) or equal volume of DMSO (−) and cell extracts immunoblotted for total LRRK2, pSer935 LRRK2, total Rab10 and pThr73-Rab10, with GAPDH as loading control. Quantifications are shown below, with P values determined by a two-tailed t test. (D) LRRK2 activity was measured as in (A) in splenocytes isolated from homozygous *Lrrk2-R1441C* mutant mice (R1441C) and their wild-type littermates (WT) 24 h after i.p. injection of 100 μg of anti-CD3 antibody (red and pink, n = 4 and 3 mice per genotype) or the same volume (100 μl) of PBS (grey and blue, n = 4 and 3 mice per genotype). Statistically significant differences are indicated with p values measured by two-way ANOVA. Data from one experiment. (E) Inflammation-induced fold change in LRRK2 activity in WT (blue, n = 3) or *Lrrk2-R1441C* (red, n = 4) splenic cells calculated as ratio between LRRK2 activities in CD3-injected and the average LRRK2 activity in PBS-injected mice of the same genotype from data shown in (D). P values calculated by a two-tailed unpaired T test are shown. The contribution of genotype, inflammation and interaction between genotype and inflammation to LRRK2 activity determined by ANCOVA are shown in Table 1. Data information: Dots show data from individual mice, with means and SEM indicated by bars and error bars, ns not significant. Source data are available online for this figure.

**Table 1. Significance of contribution of treatment, genotype and their interaction to LRRK2 variance for data shown in Fig. 3D,E.**

| Cell types | Treatment (effect of anti-CD3 injection) | Genotype (effect of R1441C mutation) | Treatment : genotype |
|---|---|---|---|
| T cells | P = 0.0104 (*) | P = 0.7016 | P = 0.8969 |
| B cells | P = 0.000145 (***) | P = 2.23E-07 (***) | P = 0.002613 (**) |
| Immat neutrophils | P = 5.47E-05 (***) | P = 9.42E-07 (***) | P = 0.00189 (**) |
| Mat neutrophils | P = 0.854 | P = 2.08E-05 (***) | P = 0.391 |
| Eosinophils | P = 0.04649 (*) | P = 0.00336 (**) | P = 0.02149 (*) |
| cDC | P = 0.386 | P = 9.48E-05 (***) | P = 0.653 |
| Immat monocytes | P = 0.0307 (*) | P = 0.0287 (*) | P = 0.0399 (*) |
| Mat monocytes | P = 0.00418 (**) | P = 0.02345 (*) | P = 0.09029 |
| Macrophages | P = 0.00136 (**) | P = 0.00786 (**) | P = 0.01111 (*) |

Statistical significance indicated by * ($0.01 < P < 0.05$), ** ($0.001 < P < 0.01$) or *** ($P < 0.001$).

inflammation was exacerbated in B cells, eosinophils and macrophages from *Lrrk2-R1441C* mice (Fig. 3E). Analysis of covariance (ANCOVA) confirmed a significant contribution of the interaction between the genotype and treatment in these cells (Table 1). Thus, R1441C mutation not only enhances baseline LRRK2 activity but also promotes its activation in the context of inflammation in some, but not all, cell types.

## In vitro stimulation of splenocytes with anti-CD3 induces LRRK2 activity and EGFP-LRRK2 expression in B cells

We next focused on inflammatory signal(s) that cause changes in LRRK2 expression and activity and asked whether anti-CD3-induced effects could be recapitulated in vitro, by culturing total splenocytes with anti-CD3/anti-CD28 antibodies to activate T cells. Such stimulation resulted in marked enrichment of B cells in the mixed splenocyte culture, and loss of most other LRRK2-expressing cell types (Fig. EV6A); therefore, we focused on B cells. We found that within mixed cultures, T-cell activation increased LRRK2 activity approximately two-fold in splenic B cells (Fig. 4A) and EGFP-LRRK2 expression by ~1.5-fold (Fig. 4B). B cells isolated from such cultures (purity >95%, Fig. EV6B) displayed increased expression of MHCII and CD69 (Fig. 4C) compared to B cells isolated from control cultures containing IL-7 only, suggesting that T-cell activation in vitro caused an activation of co-cultured B cells. This activation was accompanied by a threefold increase in *Lrrk2* mRNA level in B cells from anti-CD3/CD28-stimulated cultures (Fig. 4D), indicating that LRRK2 is likely upregulated at the transcriptional level in these cells. Remarkably, incubation of B cells isolated from *EGFP-Lrrk2*-KI mice with conditioned media collected from anti-CD3/CD28-stimulated splenocytes (Fig. 4E) led to activation of B cells (Fig. 4F) and approximately two-fold increase in EGFP-LRRK2 expression (Fig. 4G), suggesting that LRRK2 is induced in B cells by a soluble factor produced by activated T cells. Thus, soluble factors produced by activated T cells can activate B-cell expression and activity of LRRK2 ex vivo.

## IL-4 induces LRRK2 activation in B cells

To identify the soluble factor(s) enhancing LRRK2 expression, we tested the ability of neutralising antibodies against several cytokines

likely to be produced by activated T cells and present in anti-CD3/CD28-stimulated conditioned media to block LRRK2 induction. We found that antibodies against IL-4, but not IFNγ, TNF or IL-17A, significantly inhibited EGFP-LRRK2 induction by conditioned media (Fig. 5A). Furthermore, recombinant murine IL-4 was sufficient to strongly upregulate EGFP-LRRK2 expression and LRRK2-dependent Rab10 phosphorylation in B cells (Fig. 5B,C). The effect of IL-4 was much more pronounced than the effect of IFNγ, known to upregulate LRRK2 in human monocytes, human neurons and murine macrophages (Cook et al, 2017; Gardet et al, 2010; Panagiotakopoulou et al, 2020). In our system, IFNγ induced only a small increase in the level and activity of LRRK2 of B cells and was not responsible for LRRK2 upregulation by anti-CD3/CD28-conditioned media. IL-4 also induced *Lrrk2* at the transcriptional level (Fig. 5D). Furthermore, quantitative mass spectrometric analysis of LRRK2 expression in IL-4-treated B cells showed a ~2.5-fold increase in LRRK2 copy numbers (Fig. 5E). To determine whether the IL-4-dependent induction of LRRK2 expression was a result of population shift towards a subset with a higher expression of LRRK2 (such as Fo or MZ, see Fig. 2D), we combined EGFP-LRRK2 measurements with B cell subset analysis. Culturing purified B cells in homoeostatic levels of IL-7 (control) for 24 h resulted in loss of MZ markers, whereas addition of IL-4 shifted the relative abundance of B cells towards transitional CD93-high T2 state, with a concomitant reduction in T1 and Fo B cells (Fig. 5F, left). Regardless, we observed that IL-4 significantly upregulated EGFP-LRRK2 in all detectable B cell subsets (Fig. 5F, right), indicating that the IL-4-dependent induction of LRRK2 is unrelated to B cell maturation state. Immunoblotting of LRRK2, pRab10 and total Rab10, in purified mature B cells cultured with IL-4 also showed that total LRRK2 levels and LRRK2 activity were significantly increased by IL-4 (Fig. 5G). Live-cell imaging of EGFP-LRRK2 signal in B cells revealed that LRRK2 was mostly cytoplasmic in localisation after IL-4 stimulation (Fig. 5H). Thus, we identify IL-4 as a novel inducer of LRRK2 expression and kinase activity in B cells.

## Discussion

The relatively low and highly variable prevalence of pathogenic LRRK2 mutations among PD patients (Simpson et al, 2022) prompted the development of many assays to assess LRRK2 status

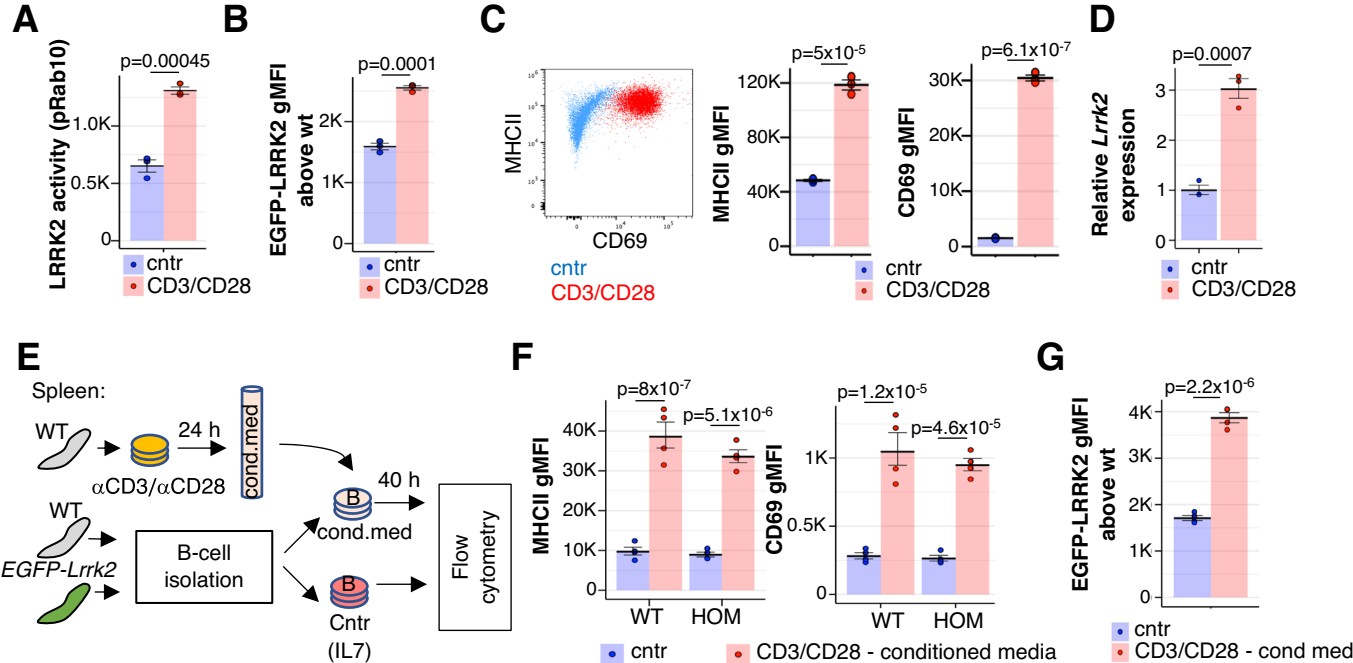

Figure 4. T-cell-dependent in vitro stimulation enhances LRRK2 expression and activity in B cells.

(A) Splenocytes from WT mice ($n = 3$) were cultured for 24 h with immobilised anti-CD3/CD28 antibodies (red), or with 1 ng/ml IL-7 as a control (blue), and LRRK2 activity measured. (B) Splenocytes from *EGFP-Lrrk2-KI* mice and their WT littermates ($n = 3$ each) were cultured as in (A) and stained with cell-type-specific markers. EGFP-LRRK2 fluorescence was measured in B cells from *EGFP-Lrrk2-KI* and WT mice, and autofluorescence of WT B cells was subtracted to determine EGFP-LRRK2 gMFI above WT background. (C) B cells were isolated from WT mouse splenocytes ($n = 3$) stimulated in vitro as in (A), and their levels of MHCII and CD69 were measured. A representative scatterplot overlay of CD69 and MHCII in B cells isolated from control (blue) or anti-CD3/CD28-induced (red) cultures is shown along with the quantification. (D) Relative mRNA levels of *Lrrk2* measured by qPCR in WT B cells ($n = 3$) from 24 h control (blue) or CD3/CD28-stimulated (red) splenocyte cultures. Bars are means of fold change relative to the control group, with *Tbp* used as a reference gene. (E) Experiment layout: splenocytes from WT mice were cultured for 24 h in vitro in the presence of immobilised anti-CD3/CD28 antibodies to stimulate T cells, after which the supernatant was collected (conditioned media) and filtered. B cells isolated from resting WT and *EGFP-Lrrk2-KI* cultures were incubated for 40 h with IL-7-supplemented media (IL-7) or with conditioned media from stimulated splenocytes before cells were harvested and measured. (F) Activation of B cells at the end of the 40 h incubation with CD3/CD28-conditioned media (red), compared with IL-7-only treated cells (cntr) as outlined in (E). is apparent from significant increases in MHCII (left panel) and CD69 (middle panel) for both WT and *EGFP-Lrrk2-KI* genotypes ($n = 4$ mice per group). (G) EGFP-LRRK2 expression in isolated B cells incubated with either IL-7 control (blue) or with CD3/CD28-conditioned media (red). Data from four individual mice. Average values from identically treated B cells obtained from four wild-type mice were used for background. Data information: (A, B, F, G): Dots show data from individual mice, with mean values and SEM depicted as bars and error bars. Statistical significance was calculated using two-tailed *t* test (for A–D, G) or two-way ANOVA (for F), and *P* values indicated above each plot. Source data are available online for this figure.

for patient stratification (Rideout et al, 2020). Most such assays utilise immunoblots, ELISA, or proteomics to examine the total level of LRRK2, its phosphorylation, or the phosphorylation of its substrates. In this study, we have independently developed a flow cytometry-based method to quantify LRRK2 activity with single-cell resolution, which builds upon previously published flow cytometry assays to assess LRRK2 kinase activity (Dhekne et al, 2023; Hakimi et al, 2011; Wallings et al, 2022). We introduced a crucial control, namely treating cells with MLi-2 prior to flow cytometry. Since the background signal of pRab10 varies between different cell types, this control is essential for the specific evaluation of LRRK2-dependent phosphorylation of Rab10, allowing for a more accurate estimate of LRRK2 kinase activity across different cell types.

We complemented the pRab10 assay with the development of the *EGFP-Lrrk2-KI* reporter mouse. Although the reporter was initially designed as a fluorescent tracker for imaging LRRK2 localisation in cells and tissues, the low expression of LRRK2,

combined with high and variable autofluorescence in intestinal tissue, precluded its use for microscopy in tissues. Even in neutrophils, which express highest level of LRRK2 among immune cells, there are less than 10,000 copies of LRRK2 per cell (Sollberger et al, 2024). However, the EGFP signal was sufficient for flow cytometry-based measurements, where background autofluorescence of each cell type was quantified and subtracted, and could be detected in isolated B cells, albeit with low signal. The reduced expression of the EGFP-LRRK2 protein and its reduced activity, possibly due to the large GFP tag destabilising the protein, indicate that caution is required while interpreting data from these mice. However, we show in two separate situations, that the EGFP-LRRK2 signal closely mirrors the LRRK2 expression data derived from proteomics of B cells from WT mice, thus supporting the use of this mouse model for evaluating relative endogenous LRRK2 expression.

Our approaches allowed us to confidently measure LRRK2 expression and activity in different types of immune cells, including

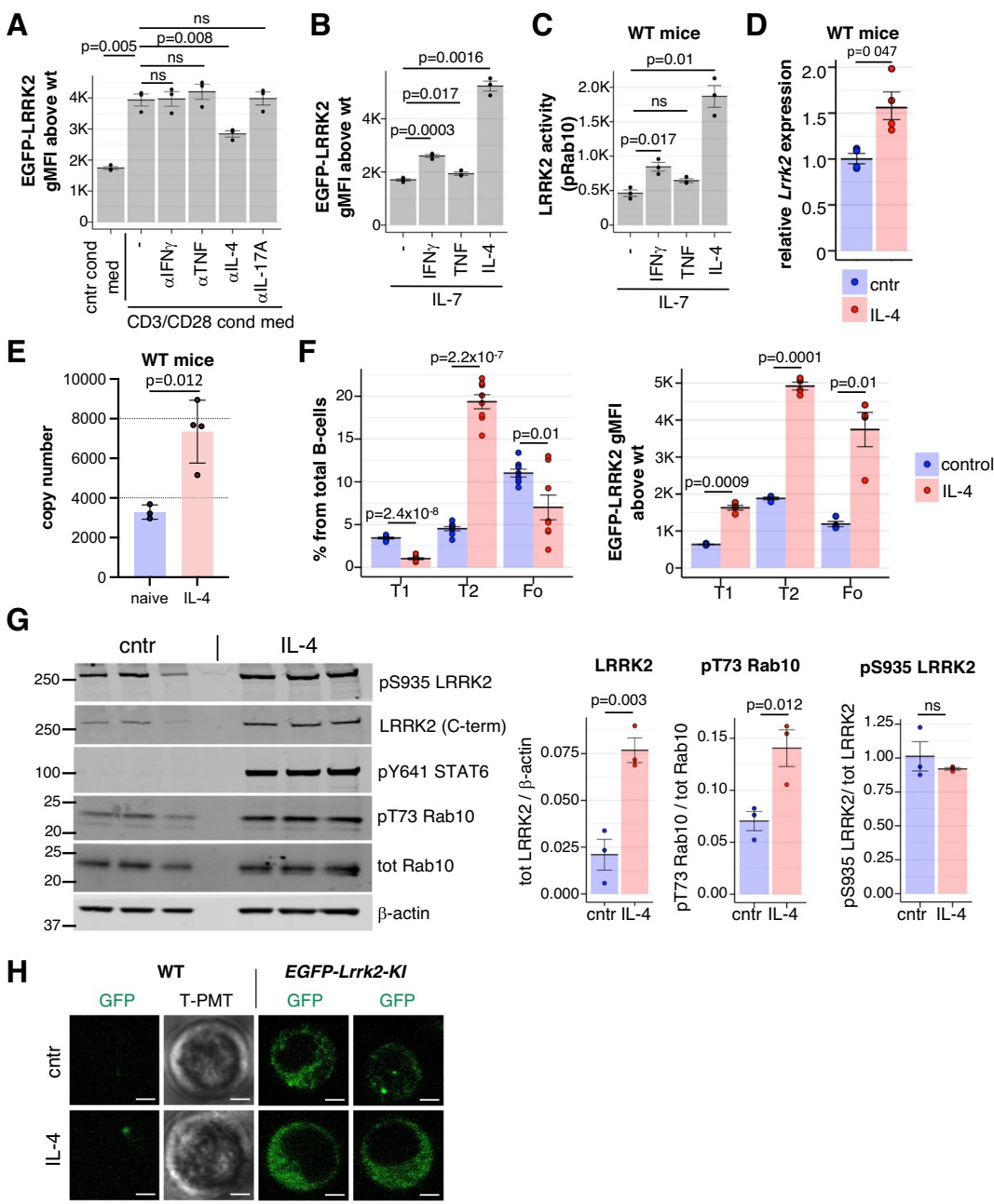

**Figure 5. IL-4 induces LRRK2 expression and activity in splenic B cells.**

(A) EGFP-LRRK2 fluorescence was measured in B cells isolated from *EGFP-Lrrk2-KI* spleens (*n* = 3 mice) and incubated for 40 h with conditioned media from splenocytes as described in Fig. 4E, supplemented where indicated with 10 μg/ml of neutralising antibodies. The autofluorescence of identically treated B cells from wild-type mice (*n* = 2) was subtracted as a background. (B) B cells isolated from *EGFP-Lrrk2-KI* spleens (*n* = 3) were cultured for 40 h in media containing 5 ng/ml IL-7 with or without 10 ng/ml IFNγ, 100 ng/ml TNF or 10 ng/ml IL-4 as indicated, and EGFP fluorescence above background measured. (C) LRRK2-dependent pRab10 gMFI (LRRK2 activity) in B cells isolated from *n* = 3 WT mice treated as in (B) was measured. (D) Relative mRNA levels of *Lrrk2* measured by qPCR in WT B cells (*n* = 4) from 24 h control (blue) or IL-4 stimulated (red) cultures. Bars are means of fold change relative to a control group, with *Tbp* used as a reference gene. (E) Graph shows the estimated copy number of LRRK2 protein per cell in lymph node naive B cells (*n* = 3 mice) or lymph node B cells stimulated for 40 h with 10 ng/ml IL-4 (data from 4 mice). (F) B cells isolated from *EGFP-Lrrk2-KI* and WT mice (4 mice per group) and cultured for 40 h with (red) or without (blue) 10 ng/ml IL-4 as in (B) were co-stained with markers for splenic B-cell subsets, and subset frequencies (left, both phenotypes pooled together) and EGFP-LRRK2 expression above background (right) were measured in T1, T2, and Fo B-cell subsets gated as in Appendix Fig. S3. (G) Immunoblot of WT splenic B cells (*n* = 3 mice) purified using positive selection and cultured for 24 h with 10 ng/ml IL-4 (IL-4) or without (cntr). Lysates from equivalent cell numbers were loaded and blotted for total LRRK2, pS935-LRRK2, pY641 STAT6, total Rab10 and pT73 Rab10. β-actin was used as a loading control. Quantifications of the total LRRK2 protein levels, LRRK2 activity towards Rab10, pS935-LRRK2 and pSTAT6 are shown. The experiment was repeated three times. (H) GFP fluorescence imaging in live B cells isolated from *EGFP-Lrrk2-KI* or WT littermate murine spleens that were cultured for 1 day with (IL-4) or without (cntr) 10 ng/ml IL-4. WT B cells shown in the left panels were additionally imaged using a transmitted light detector (T-PMT, second panels) to confirm the presence of the cells in the field of view. Scale bar = 2 μm. Images representative of three mice from three independent experiments. Data information: in (A–G), dots show values from individual mice, with means and SEM shown as bars and error bars. Statistical significance calculated by two-tailed paired *t* test (A–D, F, G) or two-tailed unpaired *t* test (E) is shown by the *P* value, ns not significant. Source data are available online for this figure.

T cells, monocytes and macrophages that have been implicated in infection control and in Crohn's disease. LRRK2 has previously been implicated in Crohn's disease through its expression in intestinal epithelial cells, specifically in Paneth cells, where it was shown to be required for anti-microbial peptide lysozyme secretion (Zhang et al, 2015). However, we and others did not find LRRK2 in murine and human Paneth cells (Sun et al, 2024; Tasegian et al, 2024). Instead, we found expression of LRRK2 in neutrophils, B cells, DCs, and monocytes, and show for the first time that eosinophils also express active LRRK2 at levels comparable to B cells and DCs. Since eosinophils have recently been implicated as key player in intestinal defence and colitis (Gurtner et al, 2023), it will be interesting to evaluate LRRK2 functions in these cells. A further interesting finding was that anti-CD3 inflammation increased LRRK2 expression and activity in B cells, immature neutrophils and monocytes, whereas, in mature neutrophils, inflammation stimulated LRRK2 activity but reduced LRRK2 levels, indicating that LRRK2 expression levels can be uncoupled from activity.

We found that inflammation robustly enhances LRRK2 expression and activity in B cells. While the role of B cells in Crohn's disease is not yet fully established (Zogorean and Wirtz, 2023), these cells are important players in several inflammatory auto-immune diseases, including SLE and multiple sclerosis (MS). LRRK2 is upregulated in B cells of patients with SLE (Zhang et al, 2019). Interestingly, the authors also found increased LRRK2 expression in published transcriptomic data (Liu et al, 2017b) from human CD19$^{hi}$/FSC$^{hi}$ B cells produced in vitro by co-culturing B cells with anti-CD3-CD28-stimulated CD4+ T cells. These CD19$^{hi}$/FSC$^{hi}$ B cells expressed activation and memory markers and higher IgG and IgM production, and are thought to represent the activated pathological CD19$^{hi}$ B cells subset found in the periphery of SLE patients (Liu et al, 2017b). Likewise, the LRRK2-upregulating activation of B cells that we observed was accompanied by elevated CD19, IgM, cell size and granularity and required activated T cells for their development, so it is possible that we are dealing with the same type of cells. However, human CD19$^{hi}$ B cells relied on the direct interaction with stimulated T cells for their development, since their numbers and ability to produce IgM and IgG was dramatically reduced by blocking either ICAM-1 that mediates such interaction, or CD40L or OX40 that are normally expressed on the surface of activated T cells. In contrast, we found that supernatant from anti-CD3/CD28-stimulated splenocytes, or Th2-type cytokine IL-4 was sufficient to upregulate LRRK2 and induce similar activation phenotype in mouse splenic B cells.

IL-4 is a pleiotropic cytokine that promotes B-cell maturation, IgG class switching and enhances production of IgG and IgA. Although it was initially described as a B-cell growth factor, on its own IL-4 is not able to promote proliferation of resting B cells (Howard et al, 1982), but it is co-stimulatory together with LPS, CD40L and antigenic stimulus. IL-4 is absolutely required for antigen-specific IgA production in the intestine after oral immunisation, and for germinal centre formation in Peyer's patches in the murine small intestine (Vajdy et al, 1995). It is interesting to note that Lrrk2-KO mice have increased levels of serum IgA (Kubo et al, 2016). Whether IL-4 induced LRRK2 in B cells is important for IgA switching in B cells, and/or for the germinal centre reaction in Peyer's patches remains to be determined. Further, IL-4 can have both pro- and anti-inflammatory roles in disease, and in one mouse model of Crohn's disease-like ileitis, a pathogenic role has been assigned to IL-4 (Bamias et al, 2005). Thus, this newly discovered IL-4–LRRK2 pathway in B cells may also contribute to autoimmune diseases.

In conclusion, we describe the utility of a LRRK2 reporter mouse model and a flow cytometric-based assay to robustly quantify LRRK2 kinase activity on a per cell basis. Using these assays, we demonstrate inflammation-driven activation of LRRK2 in vivo that is highly regulated by both cell type and tissue context. These tools enable dissection of spatial and temporal activation of LRRK2 in disease context, advancing our understanding of the critical roles different cells play in the pathological functions of LRRK2 across diseases such as SLE, PD, Crohn's, and others.

# Methods

**Reagents and tools table**

| Reagent/resource | Reference or source | Identifier or catalogue number |
|---|---|---|
| **Experimental models** | | |
| MODE-K cells (M. musculus) | Vidal et al, 1993 | |
| MODE-K, LRRK2-negative CRISPR-CAS9 clone | This paper | |
| C57BL/6J (M. musculus) | Charles River | |
| Lrrk2-KO mice (M. musculus) | Parisiadou et al, 2009 | JAX strain #012453 |
| Lrrk2-R1441C mice (M. musculus) | Jackson Laboratory | JAX strain #009347 |
| Vps35-D620N knock-in mice (M. musculus) | Jackson Laboratory | Jax strain #023409 |
| EGFP-Lrrk2 knock-in mice (M. musculus) | Taconic, this paper | |
| Healthy human volunteer blood (H. sapiens) | This paper | |
| **Recombinant DNA** | | |
| pX335 plasmid encoding spCas9n D10A nickase and a Lrrk2 antisense guide GCAGCCCTGACAGGCGCCAC(TGG) | MRC reagents | DU48900 |
| pKN7 plasmid with puromycin resistance gene and a Lrrk2 sense guide gCTCTGAAGAAGTTGATAGTC(AGG) | MRC reagents | DU48892 |
| **Antibodies** | | |
| CD45 | BD Biosciences | Clone 30-F11 |
| MHCII (I-A/I-E) | BioLegend | Clone M5/114.15.2 |
| CD11c | BioLegend | Clone N418 |
| CD11b | BioLegend | Clone M1/70 |
| Ly6G | BioLegend | Clone 1A8 |

| Reagent/resource | Reference or source | Identifier or catalogue number |
| --- | --- | --- |
| B220 | BioLegend | Clone RA3-6B2 |
| TCRb | BD Biosciences | Clone H57-597 |
| SiglecF | BD Biosciences | Clone E50-2440 |
| CD64 | BioLegend | Clone S18017D |
| Ly6C | Invitrogen | Clone HK1.6 |
| F4/80 | BioLegend | Clone BM8 |
| NK1.1 | BioLegend | Clone pk136 |
| CD3 | BioLegend | Clone 17 A2 |
| CD19 | BioLegend | Clone 6D5 |
| EpCAM | eBioscience | Clone G8.8 |
| CD93 | BioLegend | Clone AA4.1 |
| IgM | BioLegend | Clone RMM-1 |
| CD23 | BioLegend | Clone B3B4 |
| CD21/CD35 | BioLegend | Clone 7E9 |
| CD69 | Invitrogen | Clone H1.2F3 |
| LRRK2 | AbCam | Clone MJFF2 (c41-2) |
| E-cadherin | BD Biosciences | Clone 36/E-cagherin (RUO), #610182 |
| LRRK2 | NeuroMab | Clone N241A/34, #75-253 |
| DyLight™ 649 Donkey anti-rabbit IgG | BioLegend | #406406 |
| goat anti-mouse IgG Alexa Fluor Plus 488 | Invitrogen | #A32723 |
| goat anti-rabbit Alexa Fluor-568 secondary ab | Invitrogen | #A11036 |
| Biotin-CD23 | BD Pharmingen/ BD Biosciences | #553137 |
| Ultra-LEAF (Low Endotoxin, Azide-free) purified anti-mouse monoclonal anti-CD3ε antibody | Biolegend | Clone 145-2C11, # 100340 |
| Ultra-LEAF™ purified anti-mouse CD28 antibody | Biolegend | Clone 37.51, # 102116 |
| Ultra-LEAF™ purified anti-mouse IFNγ antibody | Biolegend | Clone R4-6A2, #505709 |
| Ultra-LEAF™ purified anti-mouse TNF antibody | Biolegend | Clone MP6XT22, #506332 |
| Ultra-LEAF™ purified anti-mouse IL-17A antibody | Biolegend | Clone QA20A33, # 946504 |
| Ultra-LEAF™ purified anti-mouse IL-4 antibody | Biolegend | Clone 11B11, #504122 |
| anti-pT73 Rab10 | AbCam | Clone MJF-R21-22-5, ab241060 |
| CD3 (human) | Biolegend | # 300308 |
| CD19 (human) | Biolegend | # 302230 |
| CD14 (human) | Biolegend | # 367112 |
| Goat anti-rabbit Alexa Fluor 647 secondary antibody | Thermo Fisher Scientific | # A-21245 |

| Reagent/resource | Reference or source | Identifier or catalogue number |
| --- | --- | --- |
| Rabbit anti-LRRK2 pS935 | MRC reagents | UDD2 |
| Anti-α-tubulin | Cell Signaling Technology | clone DM1A, mAb #3873 |
| Anti-total Rab10 | AbCam | #ab237703 |
| Anti-total Rab10 | nanoTools | clone 605B11, #0680/10 |
| Anti-phospho-Thr73-Rab10 | AbCam | #ab230261 |
| Anti-phospho-Tyr641-STAT6 | Cell Signaling Technology | #9361 |
| Anti-Beta Actin | Proteintech | #66009-1-Ig |
| Anti-GAPDH | Santa Cruz Biotechnology | #sc-32233 |
| Goat anti-mouse IRDye 680LT | LI-COR | #926-68020 |
| Goat anti-mouse IRDye 800CW | LI-COR | #926-32210 |
| Donkey anti-rabbit IRDye 800CW | LI-COR | #926-32213 |
| **Oligonucleotides and other sequence-based reagents** | | |
| *Lrrk2* forward primer, 5'-CAGCTTCAGAAGGGACAAGG-3' | Eurofins, this paper | |
| *Lrrk2* reverse primer, 5'-AAGGCTGCGTTCTCAGGATA-3' | Eurofins, this paper | |
| *Usp18* forward primer, 5'-TTGGGCTCCTGAGGAAACC-3' | Eurofins, this paper | |
| *Usp18* reverse primer, 5'-CGATGTTGTGTAAACCAACCAGA-3' | Eurofins, this paper | |
| *Tnf* forward primer, 5'-CTGTAGCCCACGTCGTAGC-3' | Eurofins, this paper | |
| *Tnf* reverse primer, 5'-TTGAGATCCATGCCGTTG-3' | Eurofins, this paper | |
| *Ifng* forward primer, 5'-TTACTGCCACGGCACAGTC-3' | Eurofins, this paper | |
| *Ifng* reverse primer, 5'-AGATAATCTGGCTCTGCAGG-3' | Eurofins, this paper | |
| *S100a8* forward primer, 5'-CCGTCTTCAAGACATCGTTTGA-3' | Eurofins, this paper | |
| *S100a8* reverse primer, 5'-GTAGAGGGCATGGTGATTTCCT-3' | Eurofins, this paper | |
| *Tbp* forward primer, 5'-GGGGAGCTGTGATGTGAAGT-3' | Eurofins, this paper | |
| *Tbp* reverse primer, 5'-CCAGGAAATAATTCTGGCTCAT-3' | Eurofins, this paper | |
| **Chemicals, enzymes and other reagents** | | |
| Animal-free recombinant murine IFNγ | PeproTech | #AF-315-05 |
| Recombinant murine TNF | PeproTech | # 315-01 A |
| Recombinant murine IL-4 | PeproTech | #214-14 |
| DMEM | Sigma-Aldrich | #D5671 |
| RPMI 1640 | Gibco | #11875093 |
| Foetal bovine serum | Gibco | # A5256701 |
| Penicillin–streptomycin | Sigma-Aldrich | #P4333 |
| Lipofectamine LTX with PLUS Reagent | Invitrogen/ Thermo Fisher Scientific | #15338030 |

| Reagent/resource | Reference or source | Identifier or catalogue number |
| --- | --- | --- |
| Puromycin | Sigma-Aldrich | #P9620 |
| Dulbecco's Phosphate Buffered Saline | Gibco | #14190144 |
| MLi-2 | Synthesised in house, Fell et al, 2015 | |
| Hanks' Balanced Salt Solution | Gibco | #14175095 |
| Mouse on Mouse Blocking Reagent | VectorLabs | #MKB-2213 |
| Vectashield Vibrance antifade mounting media | Vector Laboratories | #H-1700 |
| OCT compound | Agar Scientific | #AGR1180 |
| Acti-stain™ 488 phalloidin | Universal Biologicals | #PHDG1-A |
| DAPI | Invitrogen | # D1306 |
| ProLong Gold antifade reagent | Thermo Fisher Scientific | #P36930 |
| Falcon™ 70 μm nylon Cell Strainers | Corning | #352350 |
| ACK Lysing Buffer | Gibco | # A1049201 |
| EasySep Mouse Pan-B Cell Isolation Kit | StemCell Technologies | #19844A |
| Mojosort Streptavidin Nanobeads | Biolegend | #480016 |
| EasySep magnet | StemCell Technologies | #18000 |
| collagenase/dispase | Roche | #11097113001 |
| Recombinant RNase-free DNase I | Roche | #04716728001 |
| EasySep Direct Human Neutrophil Isolation Kit | StemCell Technologies | #19666 |
| Ficoll-Paque PREMIUM density gradient | GE Healthcare | #17-5442-02 |
| SepMate™ tubes | StemCell Technologies | #85450 |
| LIVE/DEAD™ Fixable Near-IR | Thermo Fisher Scientific | #L10119 |
| Ghost Dye Violet 450 | Cytek | #13-0863-T100 |
| Purified anti-mouse CD16/32, clone 93 (FC block) | BioLegend | #101302 |
| Permeabilisation buffer | Invitrogen/ Thermo Fisher Scientific | #00-8333-56 |
| Live/Dead fixable stain | Thermo Fisher Scientific | #L23105 |
| human FC block | BD Bioscience | #564219 |
| 2% paraformaldehyde | Thermo Fisher Scientific | #J19943.K2 |
| Normal Goat Serum | AbCam | #ab7481 |
| cOmplete EDTA-free protease inhibitor cocktail | Roche | #11836170001 |
| Protein Assay Dye Reagent Concentrate | Bio-Rad | #5000006 |
| NuPage LDS Sample buffer | Invitrogen | #NP0008 |
| NuPAGE 4–12% Bis–Tris Midi Gels | Thermo Fisher Scientific | #WG1403BOX |
| NuPAGE MOPS SDS running buffer | Thermo Fisher Scientific | #NP0001-02 |
| Amersham Protran Supported 0.45 mm NC nitrocellulose membrane | GE Healthcare | #10600008 |
| GFP-trap beads | Chromotek | #gta |
| 0.22 μm Spinex column | Corning | # CLS8161 |
| PureLink™ RNA Mini Kit | Invitrogen/ Thermo Fisher Scientific | #12183025 |
| PrimeScript™ reagent Kit with gDNA Eraser | Takara | #RR047A |
| TB Green® Premix Ex Taq™ II (Tli TNaseJ Plus) | Takara | #RR820L |
| Zymosan A | Sigma | #Z4250-250MG |
| S-Trap™ Mini Column | ProtiFi | |
| Invitrogen EZQ Protein Qiantification Kit | Fisher Scientific | |
| **Software** | | |
| ZEN | Zeiss | |
| OMERO with OMERO.figure (up to version 5.6) and OMERO.web up to 5.28.9. | OME | |
| SpectroFlo® software | Cytek | |
| FlowJo v10.8.1 | FlowJo LCC | |
| Image Studio Lite | Li-COR | RRID:SCR_002285 |
| CFX Maestro software | Bio-Rad | |
| Excel | Microsoft | |
| R version 4.2.0 (2022-04-22) with RStudio 2022.02.2 build 485 "Prairie Trillium" Release | | |
| **Other** | | |
| MA900 Cell Sorter | Sony Biotechnology | |
| 3-laser Cytek® Aurora spectral cytometer | Cytek | |
| BD LSR Fortessa | BD | |
| TissueLyser LT | QIAGEN | #85600 |
| Microtome | Leica Biosystems | |
| Pressure cooker (Antigen Retriever) | Aptum Biologics Ltd | |
| SLM 710 confocal microscope | Zeiss | |
| Cryolys Evolution and Precellys tubes | Bertin Technologies | |
| Odyssey CLx imaging system | Li-COR | |
| Thermocycler CFX Opus 384 | Bio-Rad Laboratories | |

Anti-mouse flow cytometry antibody staining panels:

| Antibody against | Clone | Source | a) | b) | c) | d) | e) | f) |
|---|---|---|---|---|---|---|---|---|
| CD45 | 30-F11 | BD Biosciences | + | + | + | | | + |
| MHCII (I-A/I-E) | M5/114.15.2 | BioLegend | | + | + | + | | |
| CD11c | N418 | BioLegend | + | + | + | | | |
| CD11b | M1/70 | BioLegend | + | | | | | + |
| Ly6G | 1A8 | BioLegend | + | | | | | + |
| B220 | RA3-6B2 | BioLegend | + | | | | + | + |
| TCRb | H57-597 | BD Biosciences | + | | | | | + |
| SiglecF | E50-2440 | BD Biosciences | + | + | | | | |
| CD64 | S18017D | BioLegend | + | + | | | | + |
| Ly6C | HK1.6 | Invitrogen | + | | | | | |
| F4/80 | BM8 | BioLegend | + | | | | | + |
| NK1.1 | pk136 | BioLegend | | | + | | | |
| CD3 | 17A2 | BioLegend | | | + | + | | |
| CD19 | 6D5 | BioLegend | | | + | + | + | + |
| EpCAM | G8.8 | eBioscience | | | + | | | |
| CD93 | AA4.1 | BioLegend | | | | + | | |
| IgM | RMM-1 | BioLegend | | | | + | | |
| CD23 | B3B4 | BioLegend | | | | + | | |
| CD21/CD35 | 7E9 | BioLegend | | | | + | | |
| CD69 | H1.2F3 | Invitrogen | | | | | + | |

a) For splenocytes and lamina propria cells.
b) For intraepithelial immune cells and controls (EGFP-LRRK2 levels).
c) For intraepithelial immune cells and controls (LRRK2 activity).
d) For B-cell subsets.
e) For B-cell purity and activation.
f) For peritoneal lavage cells.

## Materials availability

Materials developed in this study are available upon request but may require an MTA.

## Ethics

Mice were bred and maintained with approval by the University of Dundee ethical review committee under a UK Home Office project license (PP2719506) in compliance with UK Home Office Animals (Scientific Procedures) Act 1986 guidelines. Ethical approval for the use of human blood from healthy volunteers was obtained from the School of Medicine and Life Sciences Research Ethics Committee, University of Dundee (UOD-SMED-SLS-Staff-2024-24-37). Donors provided written consent.

## Cell lines

Murine small intestinal epithelial cell line MODE-K (Vidal et al, 1993) (kind gift from D. Kaiserlian, Lyon) was cultured in DMEM containing 10% foetal bovine serum (FBS) and 100 U/ml penicillin–streptomycin. Cells were regularly tested for mycoplasma in house. To generate

LRRK2-KO MODE-K clones, parental MODE-K cells were co-transfected with the pX335 plasmid encoding spCas9n D10A nickase and an antisense guide GCAGCCCTGACAGGCGCCAC(TGG), and the pKN7 plasmid with puromycin resistance gene and a sense guide gCTCTGAAGAAGTTGATAGTC(AGG), both targeting exon 1 of mouse LRRK2, using Lipofectamine LTX with PLUS Reagent (Invitrogen), and selected for 2 days with 5 μg/ml puromycin. Single cells from the resulting culture were sorted into 96-well plates using MA900 Cell Sorter (Sony Biotechnology) and the colonies were expanded, collected and screened for the loss of LRRK2 and Rab10 phosphorylation using immunoblotting and pRab10 flow cytometry (see below).

## Mice

Lrrk2-KO (Lrrk2$^{-/-}$) mice, JAX strain #012453 (Parisiadou et al, 2009), were from Prof. Huaibin Cai (NIH, Bethesda) and maintained on C57BL/6J background. Lrrk2-R1441C knock-in mice (Tong et al, 2009) were obtained from the Jackson Laboratory (JAX strain #009347) and maintained on C57BL/6J background. Vps35-D620N knock-in mice (Mir et al, 2018) were from the Jackson Laboratory (JAX strain #023409). All mice were maintained in a pathogen-free conditions in a standard barrier facility on a 12 h light/dark cycle at 21 °C in individually ventilated cages with sizzler-nest material and fed an R&M3 diet (Special Diet Services, UK) and filtered water ad libitum. Cages were changed at least every 2 weeks. Reporting follows the ARRIVE guidelines as closely as possible.

## Generation of EGFP-Lrrk2 knock-in mice

EGFP-LRRK2 knock-in mice were custom-made by Taconic Biosciences. The targeting vector was constructed by inserting EGFP sequence between translation initiation codon and the second codon in exon 1 of Lrrk2 (NCBI transcript NM_025730.3), and inserting a positive selection marker, PuroR, flanked by FRT sites into intron 2. The targeting vector was generated using BAC clones and transfected into C57BL/6N Tac ES cell line. After homologous recombinant clones were selected, the selection marker was removed by Flp-mediated excision, producing the constitutive KI allele encoding EGFP-LRRK2 fusion protein expressed from endogenous Lrrk2 promoter. Mice were back-crossed at least ten generations to C57BL/6J.

## MLi-2 treatment of EGFP-Lrrk2 mice

For immunoblotting analysis of EGFP-Lrrk2 mice, tissues were collected from mice injected subcutaneously with either the LRRK2-specific inhibitor MLi-2 or vehicle, 40% (w/v) (2-Hydroxypropyl)-β-cyclodextrin. Mice were injected with MLi-2 at 30 mg/kg final dose or vehicle and euthanized by cervical dislocation 2 h later. This dose and duration of treatment have been shown to effectively inhibit LRRK2 in all tissues examined including brain (Nirujogi et al, 2021).

## Anti-CD3-induced inflammation in vivo

Mice were injected intraperitoneally (i.p.) with 100 μg of Ultra-LEAF (Low Endotoxin, Azide-free) purified anti-mouse monoclonal anti-CD3ε antibody, clone 145-2C11 (Biolegend, #100340) in Dulbecco's

Phosphate Buffered Saline (PBS, Gibco) in 100 µl volume, or 100 µl of PBS and maintained for 24 h under observation before tissue harvesting. The allocation of mice into treatment groups was random. At the harvesting stage, no blinding was possible due to the apparent phenotype. No mice were excluded.

## Zymosan-induced peritonitis

Sterile acute peritonitis was induced in 15-week-old female C57BL/6 mice by intraperitoneal injection of 1 mg of zymosan A (Sigma Z4250-250MG) per mouse. Peritoneal exudate and spleens were collected at 18 h. The allocation of mice into treatment or control (equal volume of PBS) groups was random.

## Tissue H&E and immunofluorescence

Isolated mouse small intestine was flushed with HBSS, and 1.5-cm fragments of ileum were fixed in 4% Paraformaldehyde (PFA) in PBS (pH = 7.4) for 24 h at +4 °C with gentle rocking and washed three times in PBS prior paraffin perfusion and embedding into paraffin blocks. In all, 5-µm sections from paraffin-embedded tissues were deparaffinised using Histo-Clear xylene substitute, rehydrated, and either stained in Mayer's haematoxylin & eosin, or processed for immunofluorescence (IF). For IF, the antigen retrieval was performed in 10 mM Trisodium citrate containing 0.05% Tween 20, pH = 6.0, in a pressure cooker. The tissues were permeabilised for 20 min in 1% NP-40 in PBS, blocked for 1 h in PBS containing 2% Bovine Serum Albumin (BSA), 5% normal goat serum and 0.1% Triton X-100 with one drop of Mouse on Mouse Blocking Reagent (VectorLabs, #MKB-2213) per 600 µl, and stained with the primary antibodies diluted in PBS containing 2% BSA and 0.1% Triton X-100 (anti-LRRK2, clone MJFF2 (c41-2), rabbit monoclonal antibody at 1:100 dilution and anti-E-cadherin (BD Bioscences, clone 36/E-cadherin (RUO), #610182) antibody at 1:150 dilution, or anti-LRRK2, clone N241A/34, mouse monoclonal antibody at 1:200 dilution) at 4 °C overnight, followed by staining with appropriate secondary antibodies (goat anti-rabbit IgG Alexa Fluor-568, Invitrogen #A11036, and goat anti-mouse IgG Alexa Fluor Plus 488, Invitrogen #A32723, both highly cross-adsorbed to avoid any cross-species detection) diluted to 1:500 in PBS containing 2% BSA and 0.1% Triton X-100 for 1 h at room temperature. Washing was done with PBS throughout the procedure. The tissues were counterstained with 1 µg/ml DAPI and slides were mounted in Vectashield Vibrance antifade mounting media (Vector Laboratories, H-1700). Fragments of the lung were fixed at room temperature in 2% PFA in PBS (pH = 7.4) for 2–3 h, washed three times in 50 mM ammonium chloride in PBS, rinsed and incubated in 30% sucrose in PBS for 24 h at +4 °C, before embedding into OCT compound (Agar Scientific, AGR1180) on dry ice. In all, 15-µm cryosections were permeabilised for 15 min in 0.2% saponin in PBS, blocked in PBS containing 2% BSA and 0.025% saponin and incubated with anti-LRRK2 [MJFF2 (c41-2)] antibody diluted 1:100 in PBS/2% BSA/0.025% saponin overnight at +4 °C, followed by 1 h incubation with goat anti-rabbit Alexa Fluor-568 secondary ab (Invitrogen, #A11036) at 1:500 dilution with Acti-stain™ 488 phalloidin (Universal Biologicals, #PHDG1-A) at 1:75 dilution in PBS/2% BSA/0.025% saponin. Slides were counterstained with 1 µg/ml DAPI and mounted in Vectashield Vibrance antifade mounting media.

Stained tissues were imaged on a confocal Zeiss 710 microscope operated by ZEN software (Zeiss) using a 20× Plan Apochromat 0.8 NA dry objective with zoom 0.6-1. All images, including controls in which primary antibodies were omitted, were acquired using identical microscopy settings. The resulting two- or three-channel images were further processed and assembled in OMERO.figure (https://www.openmicroscopy.org/omero/figure/), identically within each experiment.

## Cell immunofluorescence

Parental MODE-K cells and LRRK2-KO MODE-K clone were seeded onto autoclaved glass coverslips placed in the wells of six-well plates, $1.6 \times 10^5$ cells per well 24 h before fixing/permeabilising them with either 4% PFA in PBS supplemented with 1% Triton X-100 for 10 min at +37 °C or with ice-cold 100% methanol for 5 min at −20 °C. Coverslips with cells were washed with PBS, blocked in PBS containing 2% BSA, 5% Normal Goat Serum and 0.1% Triton X-100 and stained overnight at +4 °C in either N241A/34 or c41-2 anti-LRRK2 antibody, diluted 1:100 or 1:200, respectively, in above blocking buffer, followed by several PBS washes and 1 h incubation with secondary antibodies (either goat anti-rabbit IgG Alexa Fluor-568 or goat anti-mouse IgG Alexa Fluor Plus 488) diluted 1:500 in blocking buffer, before counterstaining with 1 µg/ml DAPI and mounting in ProLong Gold antifade reagent, Thermo Fisher Scientific, #P36930. Staining in which primary antibodies were omitted were used as controls. For each antibody and appropriate secondary-only controls, cells were imaged on confocal Zeiss 710 microscope operated by ZEN software (Zeiss) using a 63× Plan Apochromat 1.4 NA oil objective using identical setting, and images were further processed and assembled in OMERO as above. For live GFP imaging, a suspension of B cells isolated from EGFP-Lrrk2 HOM or WT littermates using negative selection (see below) and treated for 1 day with 1 ng/ml of IL-7 with or without 10 ng/ml IL-4 was transferred into FluoroDish (WPI World Precision Instrument) and overlayed by a $12 \times 12$ mm$^2$ thin (with less than 1 mm thickness) fragment of 1% agarose in PBS equilibrated at 37 °C before imaging in GFP channel and processing as above. Wild-type cells were additionally imaged using transmitted light detector (T-PMT).

## Isolation of primary cells from mouse tissues

Splenocytes were obtained by mashing freshly isolated spleen through Falcon™ 70-µm nylon Cell Strainer in PBS containing 2% Foetal Bovine Serum (FBS), followed by erythrocyte lysis using ACK Lysing Buffer (Gibco), resuspended in RPMI 1640/10% FBS and filtered through above 70-µm Cell Strainer. For B-cell isolation, splenocytes were washed in PBS and resuspended in PBS containing 2% FBS and 2 mM EDTA before negative selection using EasySep Mouse Pan-B Cell Isolation Kit (StemCell Technologies, #19844A) following the manufacturer instructions. Alternatively, where indicated, B cells were isolated using positive selection on Mojosort Streptavidin Nanobeads (Biolegend, #480016) coupled with Biotinylated anti-CD23 antibodies (BD Pharmingen, BD biosciences, #553137), as follows: splenocytes were resuspended in FACS buffer (PBS supplemented with 2%FBS and 2 mM EDTA) up to $1 \times 10^8$ cells/ml and incubated with anti-CD23 antibody at 1:200 dilution for 10 min at +4 °C, spun down and resuspended in the fresh FACS buffer and the Mojosort Streptavidin Nanobeads, 10 beads µl per 100 µl of cell suspension containing $10^7$ cells. After

15 min incubation on ice, cells with beads were washed once in FACS buffer, resuspended in 2.5 ml of FACS buffer and isolated on an EasySep magnet (Stemcell Technologies, #18000). The magnet-bound fraction was collected, and isolation was repeated twice more to increase purity. To obtain small intestine or large intestine lamina propria (SI LP or LI LP) cells, small intestine or large intestine was excised, flushed with Hanks' Balanced Salt Solution (HBSS, Gibco) and cut longitudinally and across into ~1-cm pieces that were washed by vigorous shaking in PBS three times, collecting tissues in the 70 μm Cell Strainers between the washes, before incubating for 30 min at 37 °C in a bacterial shaker at 225 rpm in a strip buffer (PBS containing 5% FBS, 1 mM Dithiothreitol (DTT) and 1 mM EDTA), replacing the strip buffer with the fresh one 10 min after the start. Tissues were then washed three times by vigorous shaking in PBS and incubated for 45 min at 37 °C in a bacterial shaker in 10 ml of Digest buffer (RPMI 1640 containing 10% FBS, 100 U/ml penicillin–streptomycin, 2mM L-glutamine and 1 mg/ml collagenase/dispase (Roche, #11097113001) and 20 μg/ml Recombinant RNase-free DNase I (Roche, #04716728001)). Digested cell suspension was filtered through 70 μm Cell Strainer and washed twice in RPMI 1640 containing 10%FBS. Intraepithelial lymphocytes were obtained as described (James et al, 2020). Briefly, isolated small intestine was flushed with HBSS, cut as above into warm RPMI 1640 containing 10% FBS, 2mM L-glutamine, 100 U/ml penicillin–streptomycin and 1 mM DTT and incubated for 40 min at room temperature with continuous shaking. Tissue was spun down, resuspended in the same media without DTT and vortexed for 3 min before collecting the dissociated cells through 100 μm Cell Strainer, and the resuspension/vortexing/cell collection procedure was repeated two more times with the remaining tissue, combining the IEL-containing flow-through from all rounds. The cells in the flow-through were then pelleted, resuspended in 40% Percoll, overlayed over 75% Percoll and spun for 30 min at 700 × $g$, with gentle acceleration and deceleration. Cells at the interface between 40% and 75% Percoll gradient were collected and washed once with RPMI 1640 containing 10% FBS, 2mM L-glutamine and penicillin–streptomycin.

Peritoneal lavage cells were isolated as previously described (Ray and Dittel, 2010). Briefly, the mouse was euthanized by cervical dislocation, and an incision was made in the outer skin of the abdomen using scissors and forceps, carefully exposing the peritoneal lining. A 27 G needle, attached to a 5-mL syringe, was used to slowly inject 5 mL of ice-cold PBS containing 2% FBS into the peritoneal cavity. The abdomen was gently massaged to help dislodge any attached cells into the PBS solution. A 25 G needle, also connected to a 5 mL syringe, was then inserted into the peritoneum to collect the fluid, with the needle tip being gently moved to prevent blockage by fat or other tissues. Any remaining fluid was collected using a 1 mL pipette tip after making a small incision in the inner skin. The cell suspension was centrifuged at 300 × $g$ for 10 min, the supernatant was discarded, and the cell pellet was resuspended in complete media with FBS, keeping the suspension on ice.

## Cell treatment

When indicated, splenocytes were seeded in 6-well plates and incubated for 30 min with RPMI 1640 with 10% FBS containing either 100 nM MLi-2 or 0.1% DMSO. For in vitro anti-CD3/CD28 stimulation, isolated splenocytes were seeded in RPMI 1640

media containing 10% FBS, 100 U/ml penicillin–streptomycin, 2 mM L-glutamine, 1 mM sodium pyruvate, 1% non-essential amino acids, 50 μM β-mercaptoethanol, and 2.5 mM HEPES into 6-well plates pre-coated for more than 1 h with 5.4 μg of Ultra-LEAF™ purified anti-mouse CD3e antibody (clone 145-2C11, Biolegend, # 100340) and 3 μg of Ultra-LEAF™ purified anti-mouse CD28 antibody (clone 37.51, Biolegend, # 102116) per well, and incubated in standard cell culture conditions for 24 h before harvesting. As a control, cells were incubated in the above media supplemented with 1 ng/ml murine IL-7 (PeproTech, #217-17). Isolated B cells were incubated for 40 h in the above media that was diluted 1:1 with filtered control or anti-CD3/CD28-stimulated conditioned media collected from splenocytes after 24 h incubation described above or supplemented with 5 ng/ml murine IL-7. Throughout the 40 h incubation, cells were treated with 10 μg/ml of Ultra-LEAF™ purified anti-mouse IFNγ antibody (clone R4-6A2, Biolegend, #505709), 10 μg/ml of Ultra-LEAF™ purified anti-mouse TNF antibody (clone MP6XT22, Biolegend, #506322), 10 μg/ml of Ultra-LEAF™ purified anti-mouse IL-17A antibody (clone QA20A33, Biolegend, # 946504), 10 μg/ml of Ultra-LEAF™ purified anti-mouse IL-4 antibody (clone 11B11, Biolegend, #504122), 10 ng/ml animal-free recombinant murine IFNγ (Pepro-Tech, #AF-315-05), 100 ng/ml of recombinant murine TNF (PeproTech, # 315-01 A) or 10 ng/ml of recombinant murine IL-4 (PeproTech, #214-14) or left untreated as indicated.

## Human peripheral blood neutrophil and PBMC isolation

The research was approved by the East of Scotland Research Ethics Service (REC Reference 19/ES/0031), and informed consent was obtained from all participants. Neutrophils and peripheral blood mononuclear cells (PBMCs) were isolated from fresh human peripheral blood from healthy female and male donors as described previously (Fan et al, 2018). Briefly, neutrophils were isolated from human whole blood based on immuno-magnetic negative isolation technique using the EasySep Direct Human Neutrophil Isolation Kit (STEMCELL Technologies, Cat# 19666) described in detail in (Fan et al, 2020). PBMCs were isolated from whole blood using density centrifugation with Ficoll-Paque PREMIUM density gradient (GE Healthcare, Cat# 17-5442-02) and SepMate™ tubes (Stemcell, Cat# 85450) as described in detail in (Sarhan et al, 2021). After isolation, neutrophils and PBMCs from each donor were resuspended in RPMI and treated with 200 nM MLi-2 or DMSO (0.1% v/v). The cells were then subjected to the same flow cytometric workflow for pRab10 as described below.

## Flow cytometry

For EGFP-LRRK2 measurements and cell profiling, isolated cells were stained with viability dye (DAPI, LIVE/DEAD™ Fixable Near-IR (Thermo Fisher Scientific, #L10119) or Ghost Dye Violet 450 (Cytek, Cat nr. 13-0863-T100)) followed by staining with the antibody mixture (see Reagents and tools table) for identifying splenocytes and lamina propria cells; intraepithelial immune cells, B-cell subsets or for assessing B cells purity and activation, and measured on 3-laser Cytek® Aurora spectral cytometer.

For measuring LRRK2 activity, cells were stained with LIVE/DEAD™ Fixable Near-IR dye or Ghost Dye Violet 450 followed by

30 min incubation with either 200 nM MLi-2 (synthesised by Natalia Shapiro, University of Dundee) or equal amount of vehicle (0.2% DMSO) in PBS containing 2% FBS, 1 mM EDTA and FC block (Purified anti-mouse CD16/32, clone 93, BioLegend, Cat Nr101302) at 1:200 dilution at room temperature, before adding equal volume of desired antibody mixture and incubating for another 30 min. Cells were washed, fixed for 20 min with 2% PFA in PBS and washed/permeabilised in 1× Permeabilisation buffer (Invitrogen by Thermo Fisher Scientific, #00-8333-56), followed by overnight staining with anti-pT73 Rab10 antibody (clone MJF-R21-22-5, AbCam ab241060) diluted 1:500 in 1× Permeabilisation buffer with 5% Normal Donkey Serum at +4 °C and 1 h staining with DyLight™ 649 Donkey anti-rabbit IgG (BioLegend, #406406) diluted 1:500 in 1× Permeabilisation buffer, and fluorescence intensities were measured on Cytek® Aurora as above. The unmixing and compensation were performed in SpectraFlo®, and the data were further analysed using FlowJo v10.8.1.

For LRRK2 activity measurements in human peripheral blood neutrophils and PBMCs, the isolated cells were transferred into a 96-well plate and aliquoted at $2 \times 10^6$ density per condition. Cells were resuspended in 200 µl of PBS and washed twice by centrifugation at 1500 rpm for 3 min at room temperature (RT). Subsequently, the cells were stained with viability dye, Live/Dead fixable stain (Thermo Fisher Scientific, # L23105) diluted in PBS (1:1000) and incubated in the dark for 20 min at RT. Following the incubation, the cells were washed twice with PBS as described above and the cell pellet was resuspended into 25 µl of FACS buffer (1% FBS in PBS) containing human FC block (BD Bioscience, # 564219; 1:200 dilution) and either 200 nM MLi-2 (synthesised by Natalia Shpiro, University of Dundee) or an equal amount of vehicle, 0.1% DMSO. Samples were incubated in the dark for 30 min at RT. To identify CD14+ monocyte, naive B cell, and T-cell populations in PBMCs, cells were stained with a cocktail of fluorophore-conjugated cell surface markers, including CD3 (Biolegend, Cat# 300308), CD19 (Biolegend, # 302230) and CD14 (Biolegend, # 367112) diluted in FACS buffer (1:50 dilution) for further 30 min in the dark at RT. Following the incubation, the cells were washed twice with PBS (200 µl) and centrifuged at 1500 rpm for 3 min. Cells were fixed in 100 µl of 2% paraformaldehyde (PFA, Thermo Fisher Scientific, #J19943.K2) diluted in PBS at RT in the dark for 20 min. Subsequently, 100 µl of working permeabilization buffer (diluted 1:10 in MQ water, Thermo Fisher scientific, # 00-8333-56) was added to the samples followed by 3 min centrifugation at 2000 rpm at RT. Cells were washed twice with 200 µl of working permeabilization buffer as described above. Anti-pT73 Rab10 antibody (clone MJF-R21-22-5, AbCam ab241060) was diluted in working permeabilization buffer containing 5% Normal Goat Serum (NGS, Abcam, Cat# ab7481) at 1:500 concentration and cells were incubated at 4 °C overnight. Following incubations, samples were washed three times in working permeabilization buffer (by centrifugation at 2000 rpm, 3 min, RT) and stained with 50 µl of Alexa Fluor 647 goat anti-rabbit secondary antibody ((Thermo Fisher Scientific, Cat# A-21245) 1:500, diluted in working permeabilization buffer containing 5% NGS). Samples were incubated for 1 h at RT protected from light. Cells were washed with working permeabilization buffer as described above (2000 rpm, RT, 3 min) followed by 2 washes in 200 µl of PBS. Samples were resuspended in 300 µl of FACS buffer, and the data were acquired using BD LSR Fortessa with a minimum of 50,000 event acquisition per sample. The data were further analysed using FlowJo v10.8.1. LRRK2 activity in human cells was defined as:

$$\text{LRRK2 activity} = (\text{pRab10 gMFI}_{DMSO} - \text{sec ab gMFI}_{DMSO}) \\ - (\text{pRab10 gMFI}_{MLi-2} - \text{sec ab gMFI}_{MLi-2})$$

where $\text{gMFI}_{DMSO}$ or $\text{gMFI}_{MLi-2}$ are geometric mean fluorescence intensities of samples incubated with DMSO or MLi-2, respectively.

## Immunoblotting

Cell pellets were washed in PBS, lysed in iced-cold lysis buffer (50 mM Tris–HCl, pH 7.4, 1% Triton X-100, 10% (by vol) glycerol, 150 mM NaCl, 1 mM sodium orthovanadate, 50 mM sodium fluoride, 10 mM 2-glycerophosphate, 5 mM sodium pyrophosphate, 1 mg/ml microcystin-LR, and cOmplete EDTA-free protease inhibitor cocktail (Roche, #11836170001) and clarified by centrifugation at $17,000 \times g$ for 15 min at +4 °C. Frozen mouse tissues were transferred into Precellys tubes and lysed/homogenised in the above lysis buffer in the pre-chilled Cryolys Evolution at 6800 rpm, with $3 \times 20$ s cycles and $2 \times 30$ s pauses, and clarified by centrifugation at $17,000 \times g$ for 20 min. Protein concentration in soluble lysate fraction was measured by Bradford assay (Bio-Rad Protein Assay Dye Reagent Concentrate, #5000006). The samples were prepared in 1x (final concentration) lithium dodecyl sulphate sample buffer (NuPage LDS Sample buffer Invitrogen, #NP0008) and heated at 95 °C for 5 min. In total, 20 µl (for MODE-K cells) or 37 µg (for splenocytes) of protein, or the amount of lysate corresponding to $3 \times 10^5$ cells (for B cells), was loaded per well onto NuPAGE 4–12% Bis–Tris Midi Gel (Thermo Fisher Scientific, Cat# WG1403BOX) and separated under NuPAGE MOPS SDS running buffer (Thermo Fisher Scientific, # NP0001-02). Proteins were transferred onto a nitrocellulose membrane (GE Healthcare, Amersham Protran Supported 0.45 mm NC) at 90 V for 90 min on ice in transfer buffer (48 mM Tris–HCl and 39 mM glycine supplemented with 20% methanol). The membrane cut into appropriate fragments was blocked for 1 h with 5% (w/v) skimmed milk powder in TBS-T (20 mM Tris–HCl, pH 7.5, 150 mM NaCl and 0.1% (v/v) Tween 20) at room temperature. The membrane was further incubated with rabbit anti-LRRK2 pS935 (UDD2, produced in house), anti-LRRK2 C-terminus total (NeuroMab, clone N241A/34, #75-253), anti-α-tubulin (Cell Signaling Technology, clone DM1A, mAb #3873), anti-total Rab10 (either Abcam, #ab237703, or nanoTools, clone 605B11, #0680/10), anti-phospho-Thr73-Rab10 (AbCam, #ab230261), anti-phospho-Tyr641-STAT6 (Cell Signaling Technology, #9361), anti-Beta Actin (Proteintech, #66009-1-Ig) or anti-glyceraldehyde-3-phosphate dehydrogenase (GAPDH, Santa Cruz Biotechnology, #sc-32233) antibody overnight at 4 °C. Most of primary antibodies with known concentration were used at 1 µg/ml final concentration, except for total Rab10 (nanoTools) which was used at a concentration of 2 µg/ml, or diluted 1:1000, and incubated in TBS-T containing 5% BSA with exception of α-tubulin, β-actin and GAPDH antibodies that were diluted 1:5000, 1:5000 and 1:2000, respectively. After three washes with TBS-T for 10 min each, membranes were incubated with respective species-appropriate goat anti-mouse IRDye 680LT (LI-COR, #926-68020), goat anti-mouse IRDye 800CW (#926-32210) or donkey anti-rabbit IRDye 800CW (LI-COR, #926-32213) secondary antibody diluted in TBS-T (1:10,000 dilution) for 1 h at room temperature, in the dark. Membranes were washed with TBS-T three times with a 10 min incubation for each wash and kept in the dark. Protein bands

were acquired via near-infrared fluorescent detection using the Odyssey CLx imaging system and quantified using the Image Studio software.

## Immunoprecipitation

EGFP-Lrrk2-KI lung tissues were lysed/homogenised as above with the lysis buffer containing 20 mM HEPES, pH = 7.5, 150 mM NaCl, 0.5% NP-40 supplemented with 10 mM sodium pyrophosphate, 1 mM sodium orthovanadate, 0.5 μg/ml Microcystin-LR and 1× protease inhibitor cocktail. In total, 4 mg of tissue extracts were incubated with 25 μl of packed GFP-trap beads (Chromotek) for 2 h before beads were isolated from the unbound fraction on the magnet, washed three times in Wash Buffer (20 mM HEPES pH 7.5, 150 mM NaCl, 0.1% NP-40 and protease/phosphatase inhibitors), and heated in 2× LDS Sample buffer at 70 °C for 15 min. The eluted proteins were separated from beads using top speed centrifugation through a 0.22-μm Spinex column and 10% (v/v) of 2-mecraptoethanol added to eluate.

## RT-PCR

RNA was prepared from pelleted cells or frozen ileal tissues using PureLink™ RNA Mini Kit (Invitrogen by Thermo Fisher Scientific, #12183025) following the manufacturer's instructions. RNA was converted to cDNA using PrimeScript$^{RT}$ reagent Kit with gDNA Eraser (Takara, #RR047A), and used as a template for qPCR with TB Green® Premix Ex Taq™ II (Tli TNaseJ Plus) (Takara, #RR820L) and custom-made primers (Eurofins) as detailed in the Reagents tools and table.

qPCR was performed in triplicates on thermocycler CFX Opus 384 (Bio-Rad Laboratories) operated by CFX Maestro software (Bio-Rad) using cycling protocol of 30 s at 95 °C followed by 40 cycles of 5 s at 95 °C and 60 s at 60 °C, and followed by melting curve from 65 to 95 °C. The fold change in expression was calculated using $2^{-\Delta\Delta CT}$ method (Livak and Schmittgen, 2001).

## B-cell preparation for proteomics

Lymph nodes were extracted from WT C57BL/6 mice (Charles Rivers) aged between 8 and 12 weeks and were mashed in RPMI media before filtering through a 70-μm cell strainer. For generating a pure population of naive B cells, lymphocytes were incubated with FC block and then stained with CD19 FITC, CD93 APC and DAPI and CD19+ and CD93- live cells sorted by FACS. Sorted cells were washed twice with HBSS and snap frozen in liquid nitrogen and stored at −80 °C until processing for mass spectrometry. For IL-4 stimulation, lymphocytes were suspended in RPMI at a final density of 1.5 million cells/ml and stimulated for 40 h in the presence of 10 ng/ml IL−4. Stimulated cells were FC blocked and stained with CD19 APC and DAPI, and live B cells were sorted and collected as described above.

## Proteomics sample preparation

Samples were prepared for mass spectrometry as described previously but with some modifications (Sollberger et al, 2024). B-cell pellets were lysed in 400 μl lysis buffer (5% SDS, 10 mM TCEP, 50 mM TEAB) and shaken at room temperature for 5 min at 1000 rpm, followed by boiling at 95 °C for 5 min at 500 rpm.

Samples were then shaken again at RT for 5 min at 1000 rpm before being sonicated for 15 cycles of 30 s on/30 s off with a BioRuptor (Diagenode). Benzonase was added to each sample and incubated at 37 °C for 15 min to digest DNA. Samples were alkylated with the addition of iodoacetamide to a final concentration of 20 mM and incubated for 1 h in the dark at 22 °C. Protein concentration was determined using EZQ protein quantitation kit (Invitrogen) as per the manufacturer's instructions. Protein clean-up and digestion was performed using S-TRAP mini columns following the manufacturer's instructions (Protifi). Proteins were digested with trypsin at 1:20 ratio (enzyme:protein) for 2 h at 47 °C. Digested peptides were eluted from S-TRAP columns using 50 mM ammonium bicarbonate, followed by 0.2% aqueous formic acid and with a final elution using 50% aqueous acetonitrile. Eluted peptides were dried overnight before being resuspended in 40 μl 1% formic acid ready for analysis by data-independent acquisition mass spectrometry.

## Mass spectrometry data acquisition

In all, 2 μg of peptide from each sample was analysed by single-shot data-independent acquisition (DIA) as described previously (Molina-Gonzalez et al, 2023; Sollberger et al, 2024; Walgrave et al, 2023). Peptides were injected onto a nanoscale C18 reverse-phase chromatography system (UltiMate 3000 RSLC nano, Thermo Scientific) and electrosprayed into an Orbitrap Exploris 480 Mass Spectrometer (Thermo Fisher). The following liquid chromatography buffers were used: buffer A (0.1% formic acid in Milli-Q water (v/v)) and buffer B (80% acetonitrile and 0.1% formic acid in Milli-Q water (v/v)). Samples were loaded at 10 μl/min onto a trap column (100 μm × 2 cm, PepMap nanoViper C18 column, 5 μm, 100 Å, Thermo Scientific) equilibrated in 0.1% trifluoroacetic acid (TFA). The trap column was washed for 3 min at the same flow rate with 0.1% TFA then switched in-line with a Thermo Scientific, resolving C18 column (75 μm × 50 cm, PepMap RSLC C18 column, 2 μm, 100 Å). Peptides were eluted from the column at a constant flow rate of 300 nl/min with a linear gradient from 3% buffer B to 6% buffer B in 5 min, then from 6% buffer B to 35% buffer B in 115 min, and finally to 80% buffer B within 7 min. The column was then washed with 80% buffer B for 4 min and re-equilibrated in 3% buffer B for 15 min. Two blanks were run between each sample to reduce carry-over. The column was kept at a constant temperature of 50 °C.

The data were acquired using an easy spray source operated in positive mode with spray voltage at 2.445 kV, and the ion transfer tube temperature at 250 °C. The MS was operated in DIA mode. A scan cycle comprised a full MS scan (m/z range from 350 to 1650), with RF lens at 40%, AGC target set to custom, normalised AGC target at 300%, maximum injection time mode set to custom, maximum injection time at 20 ms, microscan set to 1 and source fragmentation disabled. MS survey scan was followed by MS/MS DIA scan events using the following parameters: multiplex ions set to false, collision energy mode set to stepped, collision energy type set to normalised, HCD collision energies set to 25.5, 27 and 30%, orbitrap resolution 30,000, first mass 200, RF lens 40%, AGC target set to custom, normalised AGC target 3000%, microscan set to 1 and maximum injection time 55 ms. Data for both MS scan and MS/MS DIA scan events were acquired in profile mode. The mass spectrometry proteomics data have been deposited to the

ProteomeXchange Consortium via the PRIDE partner repository with the dataset identifier PXD056827.

## Mass spectrometry data analysis

DIA raw files were analysed using Spectronaut (Biognosys) version 17. Raw mass spec files were searched against a mouse database (Swissprot Trembl November 2023) with the following parameters: directDIA, false discovery rate set to 1%, protein N-terminal acetylation and methionine oxidation were set as variable modifications and carbamidomethylation of cysteine residues was selected as a fixed modification. Estimates of protein copy numbers per cell were calculated using the proteomic ruler (Wisniewski et al, 2014).

## Statistical analysis

Data were analysed in R, GraphPad Prism or Microsoft Excel using one- or two-way ANOVA (with Tukey post hoc analysis) or *t* test as indicated. Analysis of Variance (ANCOVA) was performed in R.

# Data availability

The mass spectrometry proteomics data have been deposited to the ProteomeXchange Consortium via the PRIDE partner repository with the dataset identifier PXD056827.

The source data of this paper are collected in the following database record: biostudies:S-SCDT-10_1038-S44319-025-00473-x.

# Peer review information

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

## Acknowledgements

The project was funded by Interline Therapeutics and the UK Medical Research Council funding to MS (MC_UU_00038/7). MS is supported by the Wellcome Trust and Royal Society (Sir Henry Dale Fellowship, 206246/Z/17/Z). DRA is supported by the UK Medical Research Council (MC_UU_00018/1) and the pharmaceutical companies supporting the Division of Signal Transduction Therapy Unit (Boehringer Ingelheim, GlaxoSmithKline, and Merck KGaA). MT was supported by a PhD Studentship that was co-funded by the UK Medical Research Council and GlaxoSmithKline, and KZ by a Wellcome Trust PhD studentship (218520/Z/19/Z). EMS is funded by the Chief Scientist Office Scotland (SCAF/18/01). We are grateful for the support and helpful discussions with Dr. Hart Rardin and Dr. Yao Wong from Interline Therapeutics. We thank Dr Natalia Shapiro for producing MLi-2. We would like to acknowledge the Resource Unit at the University of Dundee for maintaining the mice and Dr. Amanpreet Singh Chawla, University of Dundee, for intraperitoneal injections, the Flow Cytometry and Cell Sorting Facility at the University of Dundee for single-cell sorting, Dundee Imaging Facility at the University of Dundee for processing tissue samples and H&E staining, Tom MaCartney for LRRK2-targeting CRISPR-Cas9 plasmids and the Swamy lab for help with processing tissues. We thank Prof. Simon Arthur, University of Dundee, for helpful advice on B cells. We also thank the volunteers who donated blood for this research. The graphical abstract was created using BioRender.

## Author contributions

**Dina Dikovskaya**: Data curation; Formal analysis; Investigation; Visualisation; Methodology; Writing—original draft; Writing—review and editing. **Rebecca Pemberton**: Data curation; Formal analysis; Investigation; Visualisation; Writing—review and editing. **Matthew Taylor**: Formal analysis; Investigation; Visualisation; Writing—review and editing. **Anna Tasegian**: Data curation; Formal analysis; Investigation; Visualisation; Writing—review and editing. **Purbasha Bhattacharya**: Data curation; Formal analysis. **Karolina Zeneviciute**: Data curation; Formal analysis. **Esther M Sammler**: Formal analysis; Supervision; Investigation; Writing—review and editing. **Andrew J M Howden**: Data curation; Formal analysis; Investigation; Writing—review and editing. **Dario R Alessi**: Conceptualisation; Supervision; Funding acquisition; Writing—review and editing. **Mahima Swamy**: Conceptualisation; Formal analysis; Supervision; Funding acquisition; Investigation; Writing—original draft; Project administration; Writing—review and editing.

Source data underlying figure panels in this paper may have individual authorship assigned. Where available, figure panel/source data authorship is listed in the following database record: biostudies:S-SCDT-10_1038-S44319-025-00473-x.

## Disclosure and competing interests statement

MS received research funding from Interline Therapeutics for this study. The authors declare no other conflicts of interest. AT is currently an employee of Amphista Therapeutics Ltd. MT is currently an employee of GlaxoSmithKline. The funders did not play a role in the conceptualisation, design, data collection, analysis, or preparation of the manuscript.

# Expanded View Figures

**Figure EV1. Immunofluorescence of mouse tissues with c41-2 anti-LRRK2 antibodies.**

(A) Sections from paraffin-embedded ileum from a C57Bl/6 J (WT, rows 1 and 3) or a *Lrrk2*$^{-/-}$ mouse (rows 2 and 4) were co-stained with rabbit c41-2 anti-LRRK2 ab (clone MJFF2), red, and mouse anti-E-cadherin ab (green), followed by secondary anti-rabbit Alexa Fluor-568 (AF568) and anti-mouse Alexa Fluor-488 (AF488) antibodies, counterstained with DAPI (blue) and imaged by confocal microscopy (panels 1 and 2 from left). Staining that did not include primary antibodies used as a control (panels 3 and 4) was processed, imaged and adjusted in in the same way. Single optical sections are shown. Panels 2 and 4 depict areas with Peyer's Patches. Top two rows show overlay of all staining, and the bottom two rows show only c41-2 channel from the same images in black and white. Scale bar = 100 μm. Representative of at least four similar experiments. (B) Sections from frozen lung tissues from WT (top) or *Lrrk2*$^{-/-}$ (bottom) mice were stained with c41-2 anti-LRRK2 ab (clone MJFF2) (red on overlay) followed with Alexa Fluor-568 fluorescent secondary ab (panels 1 and 2), or with secondary ab only (panels 3 and 4). Sections were counterstained by DAPI (blue) and Phalloidin-488 (not shown). Confocal images were acquired and processed identically. Panels 1 and 3 show c41-2 and DAPI overlay, panels 2 and 4 show c42-1 channel only in black and white. Scale bar = 50 μm. Images were processed and assembled in OMERO.

▶

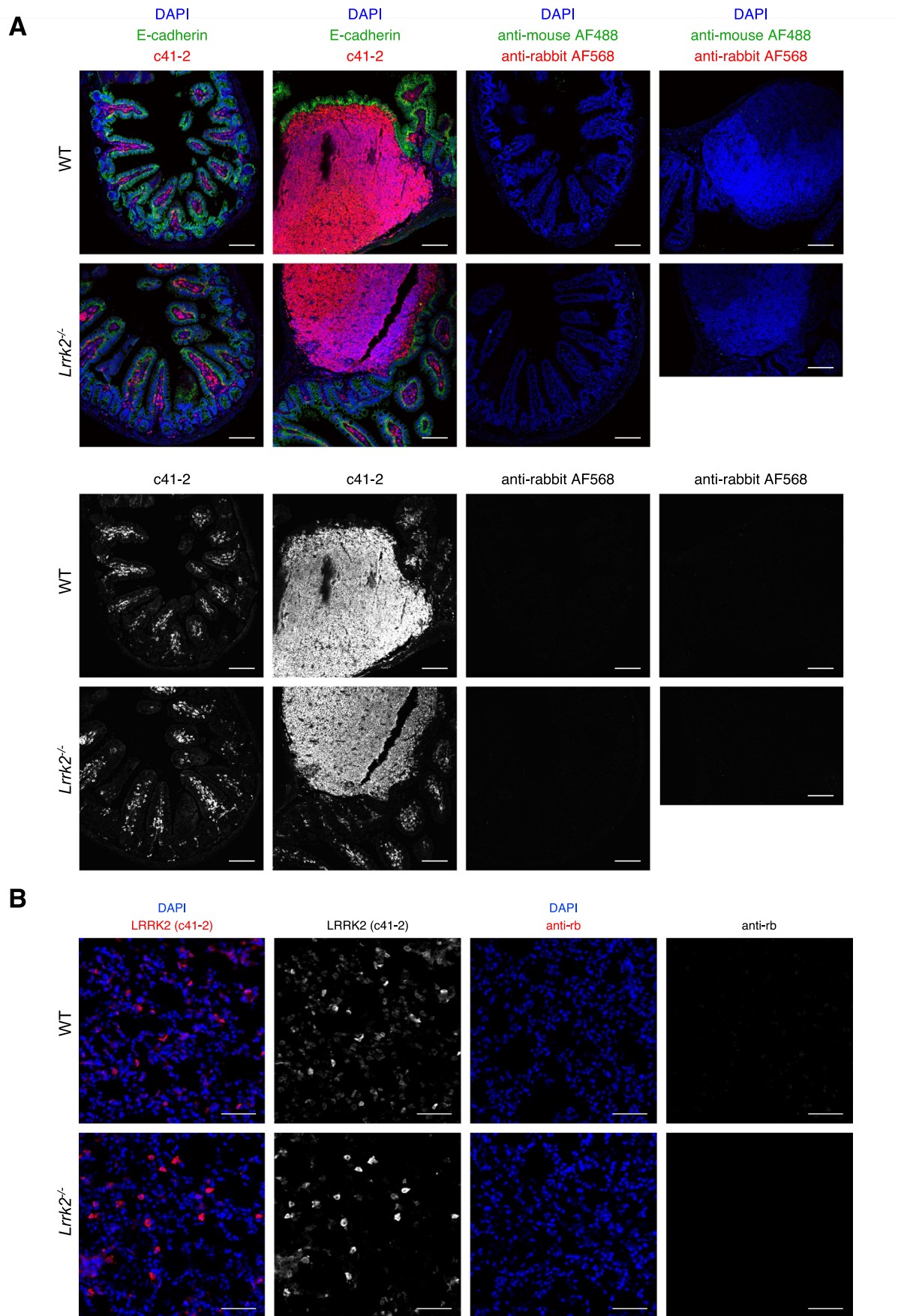

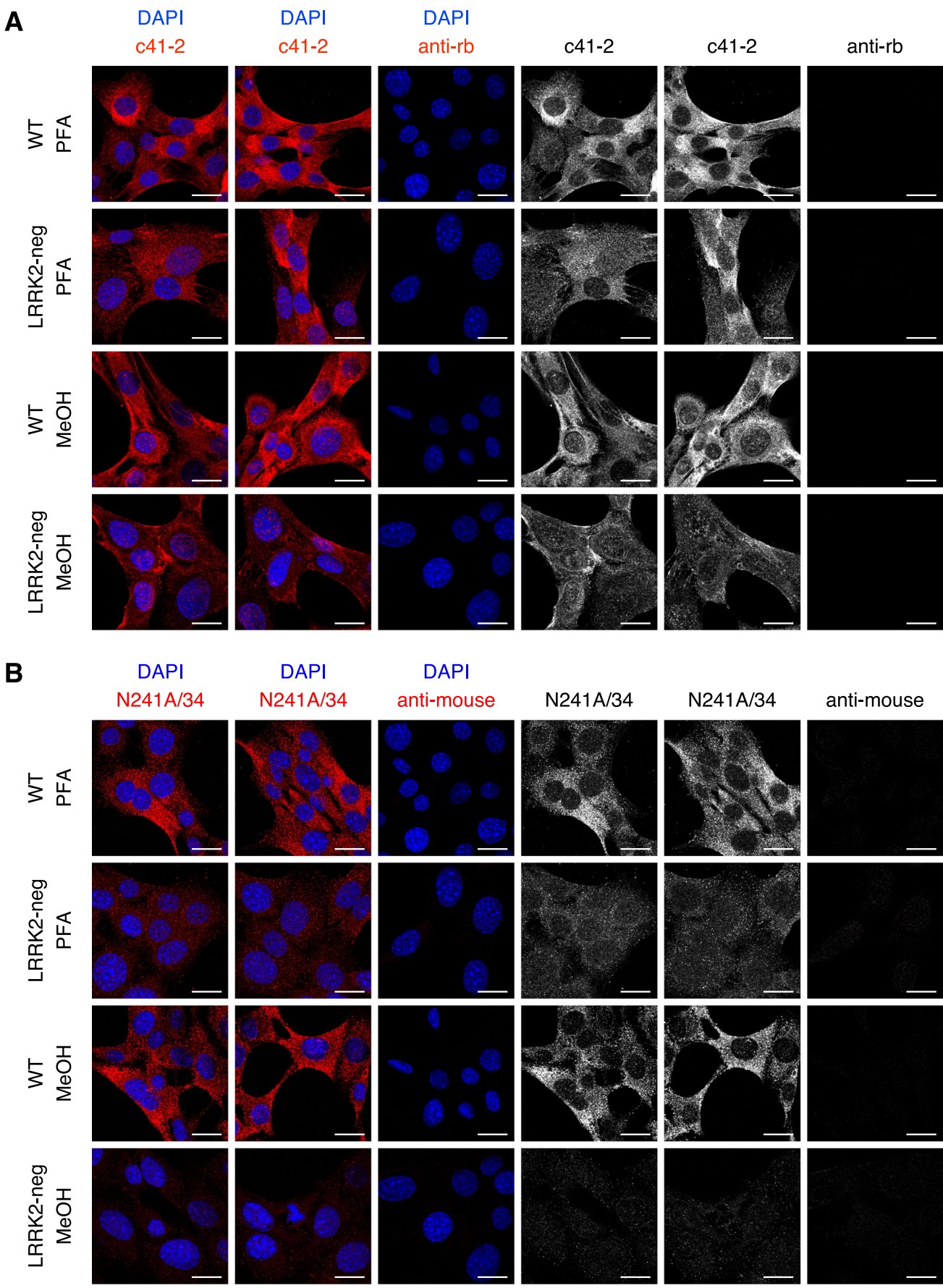

◄ **Figure EV2.  LRRK2 immunofluorescence in mouse MODE-K cell line.**

Parental MODE-K cells (WT) and the LRRK2-deficient clone (LRRK2-neg) generated by CRISPR-Cas9 knock-out of LRRK2 were grown on coverslips, fixed/permeabilised in either 4% PFA / 1% Triton X-100 (top two rows, PFA) or Methanol (bottom two rows, MeOH), and stained with either c41-2 (clone MJFF2) (**A**) or clone N241A/34 (**B**) anti-LRRK2 antibodies followed by an appropriate fluorescently-labelled secondary ab (red on overlay) and DAPI (blue on overlay). Cells were imaged on confocal microscope and images processed in OMERO. Anti-rabbit (anti-rb) or anti-mouse secondary-only controls (panels 3 and 6) were imaged and processed identically to the full-stained samples. The overlays are shown in panels 1-3, and panels 4-6 depict LRRK2-only staining in black and white. Scale bar = 20 μm.

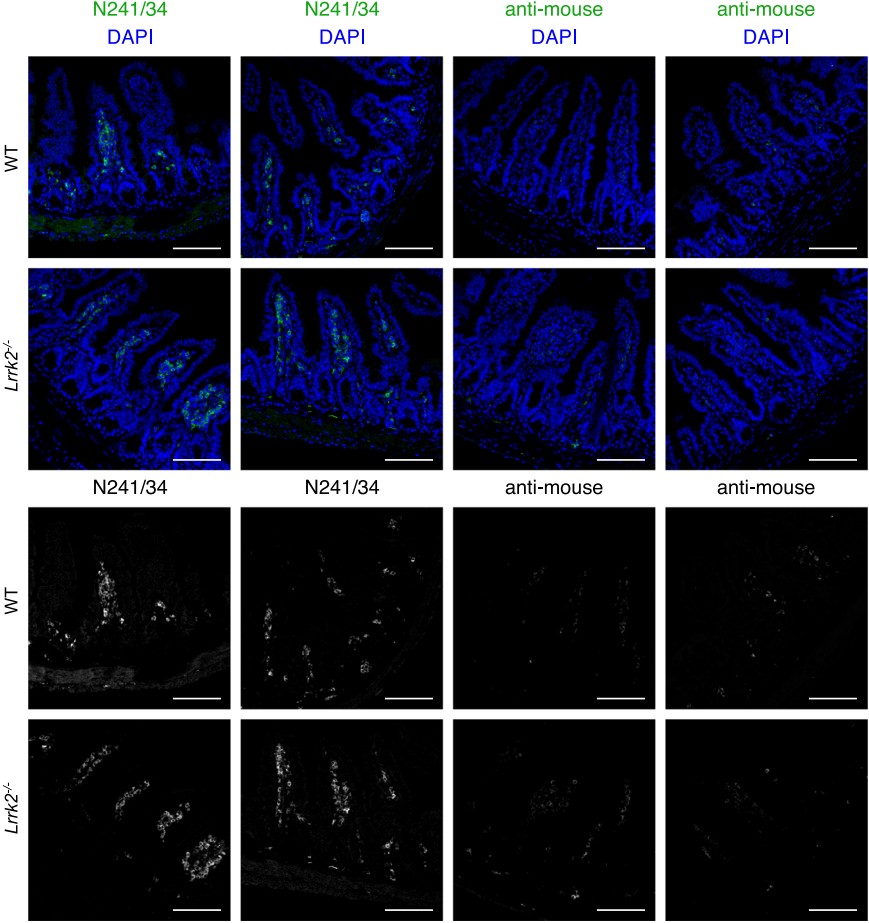

**Figure EV3. Tissue immunofluorescence with anti-LRRK2 antibody N241A/34.**

Sections of paraffin-embedded ileums from a *Lrrk2*$^{-/-}$ mouse (rows 2 and 4) and its WT littermate (rows 1 and 3) were stained with N241A/34 antibodies followed by fluorescent secondary ab (panels 1 and 2) or with secondary ab only (panels 3 and 4), (green on overlay) and counterstained with DAPI (blue on overlay). Tissues were imaged on confocal microscope and processed in OMERO. Top two rows show overlay, bottom two rows show LRRK2-only staining in black and white. Scale bar = 100 μm.

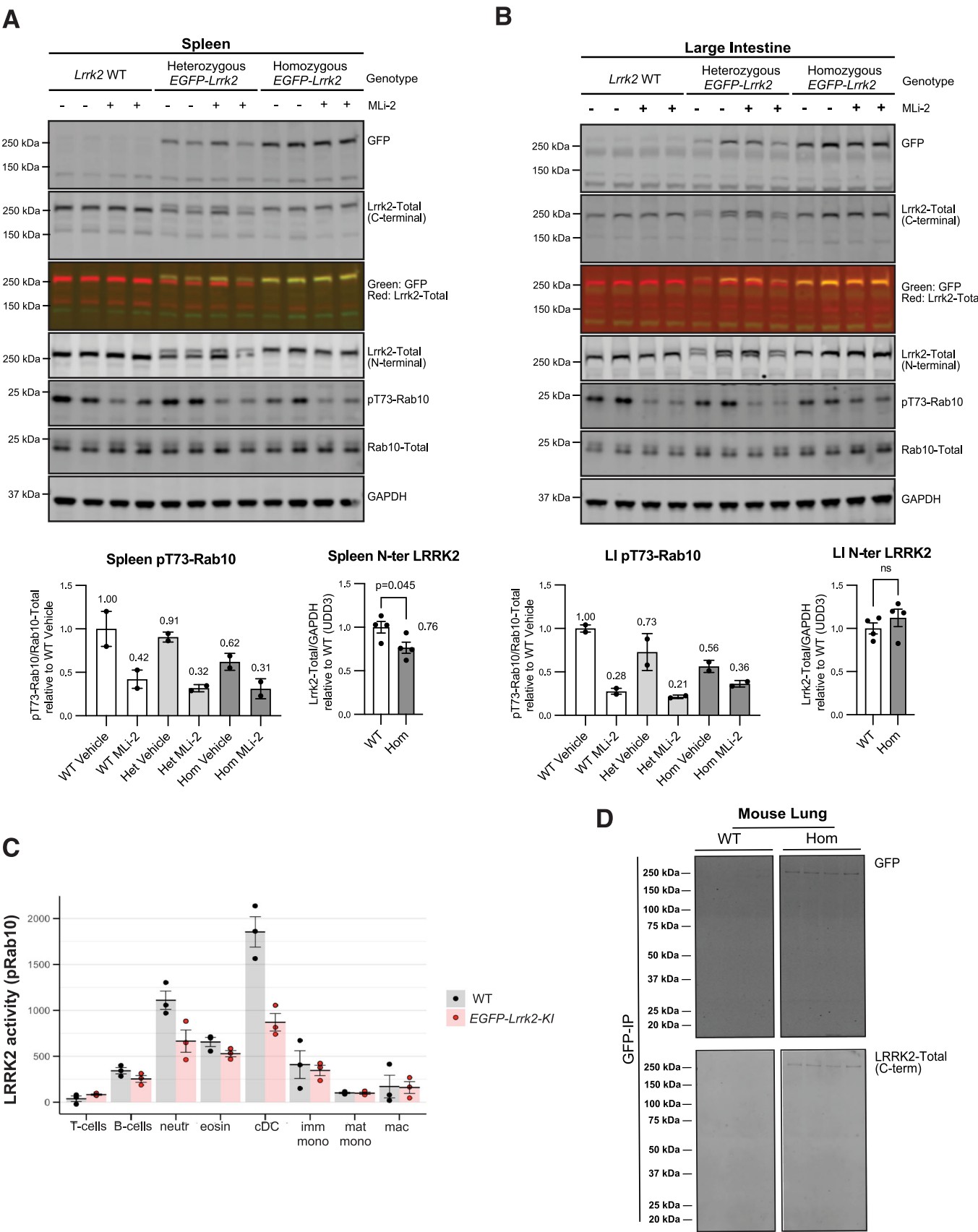

◀ **Figure EV4. Analysis of the LRRK2 pathway in tissues from *EGFP-Lrrk2-KI* mouse model.**

(A, B) 3-month-old WT, heterozygous and homozygous *EGFP-Lrrk2-KI* mice ($n = 2$ each) were treated with or without 30 mg/kg MLi-2 subcutaneously for 2 h prior to culling. Mouse tissues were immediately extracted and snap frozen in liquid nitrogen. Frozen tissues were lysed using Cryolys Evolution and between 20 to 30 μg of whole tissue lysate of either spleen (A) or large intestine (B) subjected to immunoblot analysis. Quantification of LRRK2-substrate phosphorylation of pT73 Rab10 relative to total levels ($n = 2$), and total LRRK2 levels relative to GAPDH for N-terminal antibodies ($n = 4$) are shown as mean ± SEM for each tissue. Each lane indicated sample derived from a different mouse tissue. Statistical significance was assessed by two-tailed unpaired $T$ test. ns = not significant. (C) LRRK2 activity (pRab10) was measured in splenocytes from homozygous *EGFP-Lrrk2-KI* mice (EGFP-LRRK2-KI, red, $n = 3$) or their wild-type littermates (WT, black, $n = 3$) and displayed as in Fig. 1D. Dots show measurements in individual mice, bars with error bars show means ± SEM. (D) Mouse lung tissues derived from homozygous genotypes of Lrrk2-WT ($n = 4$) and *EGFP-Lrrk2-KI* ($n = 4$) mice were homogenised using Cryolys Evolution in 0.5% NP-40 detergent lysis buffer. 4 mg of lung tissue lysate was subjected to a GFP-IP at 4 °C for 2 h. 10% of the IP eluate was subjected to immunoblot analysis using antibodies indicated.

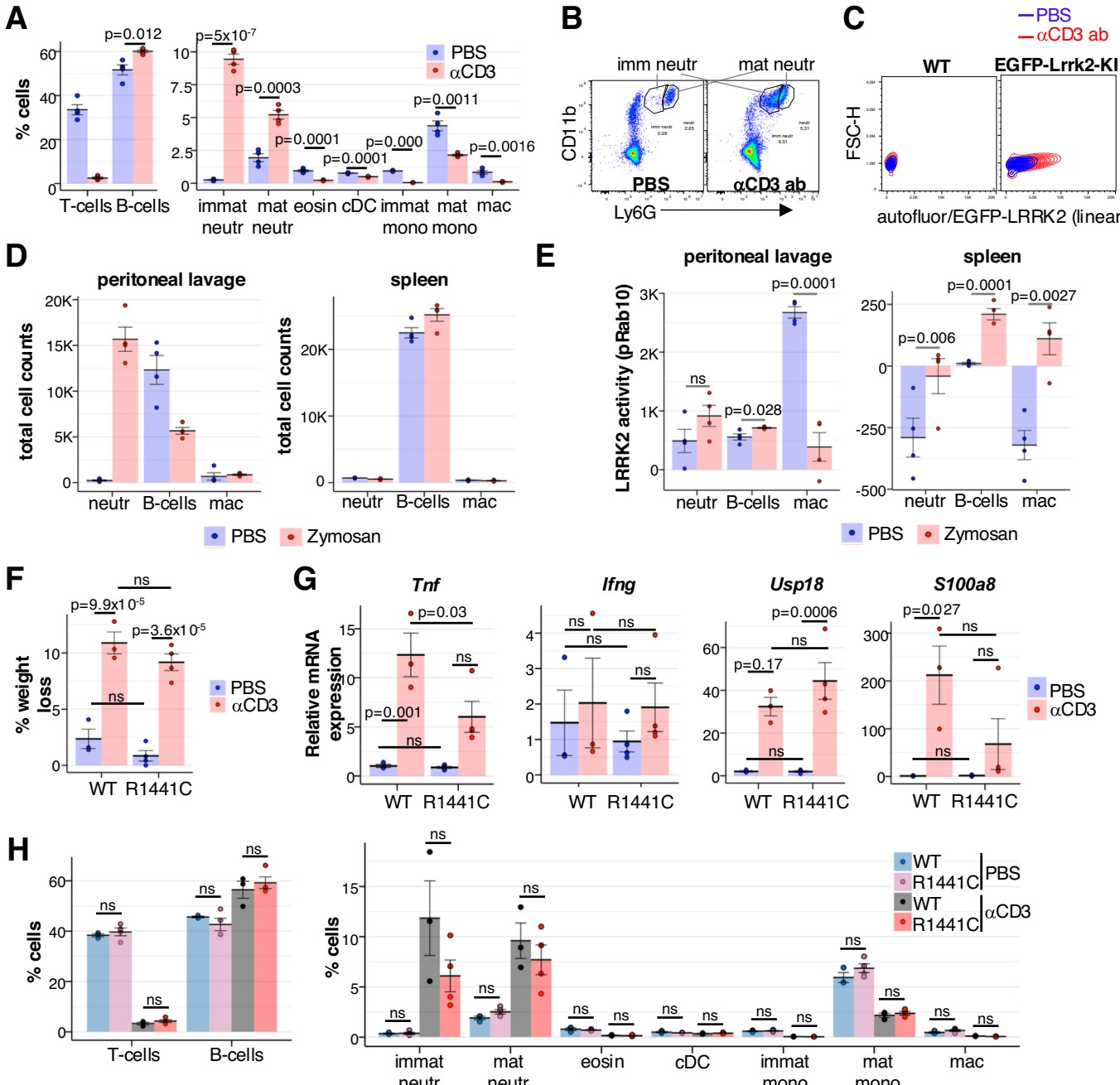

**Figure EV5. Effects of inflammation on LRRK2 activity in WT and Lrrk2-R1441C mutant mice.**

(A) Cellular composition of mouse spleen ($n = 4$ mice) 24 h after i.p. injection of anti-CD3 (αCD3, red) or PBS (blue). Frequencies of indicated cell types as a percent of single live CD45$^+$ cells is shown. Statistical analysis was not applied to T cells, since strong reduction in the number of T cells is most likely due to incomplete identification of T cells resulting from anti-CD3 induced internalisation of the TCR. (B) Distinction between mature and immature neutrophils by Ly6G level from cells in A. Note a strong expansion of the CD11b$^{hi}$/Ly6G$^{int}$ immature neutrophil subset in anti-CD3 ab injected sample. (C) EGFP-LRRK2 fluorescence (for *EGFP-Lrrk2-KI* sample) or autofluorescence measured in the same channel (for matching WT control) in B cells among splenocytes from WT (left panel) or EGFP-Lrrk2-KI (right panel) mice 24 h after anti-CD3 (red) or PBS (blue) injection, plotted on a linear scale against FSC-H. (D, E) WT mice were i.p. injected with either 1 mg/mouse Zymosan A (red, $n = 4$) or vehicle (PBS, blue, $n = 4$). 18 h later the total numbers of neutrophils, B cells and macrophages (D) and their LRRK2 activity (E) were measured in peritoneal lavage (left) and spleens (right) and displayed as in A. (F). Body weights before and 24 h after i.p. injection of anti-CD3 antibody (αCD3, red, $n = 3$ and 4) or equal volume of PBS (PBS, blue, $n = 3$ and 4) were measured for wild-type (WT) and *Lrrk2-R1441C* knock-in (R1441C) mice, respectively, and % weight loss was calculated. (G) Relative expression of *Tnf*, *Ifng*, *Usp18* and *S100a8* mRNA was measured by qPCR in ileums isolated from *Lrrk2-R1441C* mice (R1441C) and their wild-type littermates (WT) 24 h after anti-CD3 (red, $n = 4$ and 3) or PBS (blue, $n = 4$ and 3) injection. Graphs show fold changes relative to PBS-injected WT mice, with *Tbp* as a reference gene. (H). Frequencies of indicated cell types among live single CD45+ splenocytes obtained from WT or R1441C mice 24 h after anti-CD3 (grey, $n = 3$, or red, $n = 4$) or PBS (purple, $n = 3$, or blue, $n = 4$) injection are displayed. Note that there was no statistically significant difference between two genotypes. Data information: (A, E, D–H): Dots depict values in individual mice, bars show means, and error bars represent SEM. Statistical significance calculated by one-way ANOVA (A), 2-way ANOVA (F–H) or 2-tailed T test (E) is shown as P values or ns = not significant.

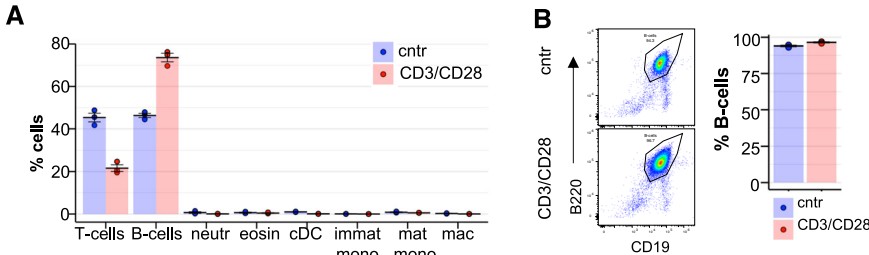

**Figure EV6.   In vitro B-cell stimulation by T-cell-dependent factors.**

(A) Frequencies of indicated cell types in total splenocyte cultures 24 h after in vitro incubation with immobilised anti-CD3/CD28 antibodies (CD3/CD28, red, $n = 3$) or in control media supplemented with IL-7 (cntr, blue, $n = 3$) were quantified as % from alive single CD45+ cells within harvested cell suspensions. (B) Purity of B cells isolated from mixed splenocyte cultures after 24 h incubation with immobilised anti-CD3/CD28 antibodies (CD3/CD28, bottom left panel, red on the plot) or in control media supplemented with IL-7 (cntr, top left panel, blue on plot). B cells were purified using negative selection kit and defined as CD19+/B220+ cells (left panels). The quantification of the purity as % from single live cells is shown on the right (3 per group). Data information: (A, B) Values from individual mice (dots), means (bars) and SEM (error bars) are shown.

