## [Peer Review File · EMBO Reports]

Inflammation and IL-4 regulate Parkinson's and Crohn's disease associated kinase LRRK2

Dina Dikovskaya, Rebecca Pemberton, Matthew Taylor, Anna Tasegian, Purbasha Bhattacharya, Karolina Zeneviciute, Esther Sammler, Andrew Howden, Dario Alessi, and Mahima Swamy

Corresponding author(s): Mahima Swamy (m.swamy@dundee.ac.uk) , Dina Dikovskaya (d.dikovskaya@dundee.ac.uk)

Review Timeline:

Transfer Date:	14th Aug 24
Editorial Decision:	20th Aug 24
Revision Received:	20th Jan 25
Editorial Decision:	26th Feb 25
Revision Received:	17th Apr 25
Accepted:	30th Apr 25

Editor: Achim Breiling

Transaction Report: This manuscript was transferred to EMBO reports following peer review at Review Commons.

**Review
COMMONS**

Revision Plan

Manuscript number: RC-2024-02505

Corresponding author(s): Mahima Swamy

1. General Statements

LRRK2 is the subject of intense research as polymorphisms in LRRK2 have been associated with increased risk for a variety of diseases, including Parkinson's, Crohn's, and Systemic Lupus Erythematosus, whilst potentially providing protection against certain infections. These diseases involve immune cells, yet the expression and function of LRRK2 in immune cells through the body has not yet been systematically evaluated. Moreover, previous analyses of LRRK2 expression have been confounded by the fact that many LRRK2 antibodies are non-specific in immunofluorescence and flow cytometry. Hence, we established new tools- a flow cytometric assay for a LRRK2 substrate, and an EGFP-Lrrk2 knockin mouse model- to investigate the expression and activity of LRRK2 in immune cells. We used these tools to evaluate LRRK2 status in mouse spleen and intestine as well as in human blood.

We note that all three reviewers found the tools we have developed to be beneficial to the field, the findings of this study as highly important. Reviewer 1 states that the study is descriptive, however we think that there is great value of describing where and when LRRK2 is expressed and active, and therefore we do not consider this as a negative. Indeed, as reviewers 2 and 3 point out, the value lies in defining the complexity of LRRK2 expression and regulation in tissues. Our hypothesis-free unbiased analysis of LRRK2 activity led us to the novel finding that IL-4 is a strong inducer of LRRK2 in B-cells.

2. Description of the planned revisions

In response to the reviewers' comments, the following experiments are planned:

Reviewer 1: The anti-CD3 strategy to induce inflammation is very different from physiological inflammation such as sepsis and LPS stimulation, so experiments with other stimuli could be important here to contribute to the message of inflammatory trigger of LRRK2 activation and decoupling of cell type.

We thank the reviewer for this suggestion. We used the anti-CD3 model as it also causes intestinal inflammation, and mimics T-cell cytokine storms that happens in many diseases. However, for the revisions we will also test another model of inflammation as suggested, such as LPS stimulation, to measure how inflammation affects LRRK2 expression and activity.

Reviewer 1: The authors generated a new mouse KI mouse expressing EGFP-LRRK2 and show data the levels of LRRK2 expression are reduced in tissues at different degrees and established a flow cytometry assay to measure LRRK2 expression by monitoring the GFP signal.[...]. If data using this mouse is included, then microscopy should be included to complement the flow cytometry data.

We tried but failed to visualize the EGFP-LRRK2 signal using fluorescence microscopy in the tissue. This is most likely due to the low expression of LRRK2 (proteomics data suggests that even neutrophils express less than 9000 copies), confounded further by the high background autofluorescence in tissues, especially in the gut. We now explain the lack of tissue images from the EGFP-LRRK2 mice in the text. However, we can visualize the EGFP-LRRK2 in B cells, and we will provide these images in a revised version of the manuscript.

Revision Plan

Reviewer 2: Since there are no good antibodies for IF/IHC as pointed by the authors, the GFP-Lrrk2 mouse gives the opportunity to check endogenous LRRK2 localization, i.e. in cells untreated or treated with IL-4 or other cytokines. Also, does endogenous GFP-LRRK2 accumulate into filaments/puncta upon MLI2 inhibition? The relocalization into filaments of inhibited LRRK2 has been observed in overexpression but not under endogenous expression. This analysis would be interesting also in light of the observed side effect of type-I inhibitors.

We will attempt a super-resolution microscopy using Airyscan with isolated B-cells treated with cytokine and/or LRRK2 inhibitor to address this question.

3. Description of the revisions that have already been incorporated in the transferred manuscript

Please insert a point-by-point reply describing the revisions that were already carried out and included in the transferred manuscript. If no revisions have been carried out yet, please leave this section empty.

Reviewer 1

The authors generated a new mouse KI mouse expressing EGFP-LRRK2 and show data the levels of LRRK2 expression are reduced in tissues at different degrees and established a flow cytometry assay to measure LRRK2 expression by monitoring the GFP signal. Interestingly they found that expression does not correlate with activity (as measured by phospho-Rabs). I suggest taking this part out as it breaks the flow of the paper. If data using this mouse is included, then microscopy should be included to complement the flow cytometry data.

In response to the comment about adding tissue microscopy we have added the following paragraph to the discussion:

“We complemented the pRab10 assay with the development of the EGFP-Lrrk2-KI reporter mouse. Although the reporter was initially designed as a fluorescent tracker for imaging LRRK2 localisation in cells and tissues, the low expression of LRRK2, combined with high and variable autofluorescence in tissues precluded its use for microscopy. Even in neutrophils, which express highest level of LRRK2 among immune cells, there are less than 9000 copies of LRRK2 per cell (Sollberger et al, 2024), making it difficult to identify localization. However, the EGFP signal was sufficient for flow cytometry-based measurements, where background autofluorescence of each cell type was taken into account and subtracted.”

Reviewer 2

1) Figure 5. The authors need to label more clearly the graphs referring to wt mice versus GFP-Lrrk2 KI mice.

We have now labelled the panels referring to the WT mice only with “WT mice”, to distinguish them from the other panels that incorporate data from both EGFP-Lrrk2 mice and their WT littermates used as a background.

2) They should also replace GFP-LRRK2 with GFP-Lrrk2 since they edited the endogenous murine gene.

Thank you, we have corrected it, and also the other mouse genotypes.

3) In the material and methods MLI-2 administration in mice is indicated at 60 mg/kg for 2 hr whereas in suppl. figure 5 the indicated dose is 30 mg/kg. Please correct with the actual dose used.

Revision Plan

Thank you, we have corrected the mistake.

Reviewer 3

P8 : the authors state that their results indicate 'that the effects of LRRK2-R1441C mutation and inflammation on LRRK2 activity represent two different parallel pathways'. This seems like an overinterpretation as pathway suggests the presence of additional partners in the pathway while R1441C is a LRRK2 intrinsic modification. The results can equally be explained by synergistic effects between both activation mechanisms (mutant and inflammation).

We agree with the reviewer, and have added this into the text. The sentence now reads “suggesting that the LRRK2-R1441C mutation and inflammation have different impacts on LRRK2 activity, either in parallel or in synergy.”

Methods and experiment descriptions in results : the authors appear to use the terms anti-CD3 stimulation and CD3 stimulation interchangeably, although it is not always clear in the text that these are synonymous. This should be clarified.

We thank reviewer for pointing out this error on our part. We have made the necessary changes to always refer to the stimulation as anti-CD3.

The authors mention in the discussion that they 'show for the first time that eosinophils also express active LRRK2 at levels comparable to B-cells and DCs.' The relevance of this finding should be further developed. Why is this important?

We thank the reviewer for this point. We don't know how LRRK2 is important in these cells. However, as the role of LRRK2 in eosinophils and neutrophils has not yet been explored and both cell types play important roles in IBD, we think it is important to point out. We have now added a sentence to the discussion highlighting the importance of eosinophils in IBD. “Since eosinophils have recently been implicated as key player in intestinal defense and colitis(Gurtner *et al*, 2022), it will be interesting to evaluate LRRK2 functions in these cells.”

Figure 5G, for the graphs indicating LRRK2 activity and LRRK2 phosphorylation, the specific measures should be specified in the graph titles to avoid any ambiguity (pT73-Rab10, pS935-LRRK2).

We have added the specifications to the new version of the figure.

Suppl figure 1 : please specify the figure label and abbreviation AF568 in the legend.

Suppl figure 2 : please specify the figure label and abbreviation anti-rb in the legend

Thank you, we added the abbreviations to the legends. The Figure labels for both figures have been already included at the top of figure legends.

4. Description of analyses that authors prefer not to carry out

Please include a point-by-point response explaining why some of the requested data or additional analyses might not be necessary or cannot be provided within the scope of a revision. This can be due to time or resource limitations or in case of disagreement about the necessity of such additional data given the scope of the study. Please leave empty if not applicable.

Reviewer 1

Revision Plan

The flow cytometry assay of the first part is a great technical challenge and represents the establishment of a potentially very useful tool for the field. It would have been important to test other organs, either as controls or for example because of their relevance e.g. lungs.

We thank the reviewer for pointing out that the pRab10 assay would be useful to apply to other organs too. Since we are interested in the role of LRRK2 in IBD, we had focused on applying the pRab10 assay on intestinal tissue, with spleens also analysed as major lymphoid organ and a source of immune cells that can translocate to the gut in inflammation. We hope that the publication of this method would allow other researchers to analyse other tissues in the future.

Reviewer 2:

The discovery of IL-4 as a Lrrk2 activator in B cells is a very interesting and novel finding. The authors could take advantage of the GFP tag to investigate LRRK2 interactome upon IL-4 stimulation (optional). Also, is the signaling downstream of IL-4 attenuated in Lrrk2 KO cells?

We thank the reviewer for these interesting suggestions. The role of LRRK2 in IL-4 activated B-cells is currently under active research in the lab. We hope to address these questions in a future manuscript.

Reviewer 3

One major observation in this paper is that LRRK2 is not detected in gut epithelial cells as previously has been reported. It would be useful to comment on any differences between the presented protocol and the previous reports, in particular relating to the antigen retrieval step. In order to reinforce the finding, it would be useful to include in situ hybridization data that could further strengthen the observations of which cellular subtypes express LRRK2 and which do not. Indeed, while the KO control shows that there is an unacceptable high non-specific staining, it does not prove absence of expression. Also, can any conclusions be made about expression of LRRK2 in neural cells of the gut? This important information on LRRK2 detection in gut should be mentioned in the abstract and highlighted in the discussion.

We thank the reviewer for pointing this out. In fact, we think the observation that LRRK2 is not detected in epithelial cells is so important that we have a separate manuscript exploring this point. Please see 1. Tasegian, A. *et al.* <https://doi.org/10.1101/2024.03.07.582590> (2024).

In this manuscript we have explored the expression of LRRK2 in human and murine intestinal epithelial cells using qPCR. Although we do not have in situ hybridization data, we believe that using both the EGFP-LRRK2 and the pRab10 flow cytometry, as well as qPCR and proteomics on selected cell types, corroborates our findings on the cell types that express LRRK2. We did not analyse LRRK2 expression in the neural cells of the gut, as the focus was on the immune cells, however we hope that others will use the tools developed here to explore this further.

Dear Dr. Swamy

Thank you for the transfer of your research manuscript from Review Commons to EMBO reports. I now went through your manuscript, the referee reports from Review Commons (attached again below) and your revision plan. The referees have several comments, concerns, and suggestions to improve the manuscript, indicating that a major revision of the manuscript is necessary to allow publication of the study.

Going through your revision plan, it seems that the referee points will be adequately addressed during revision. I thus invite you to revise your manuscript accordingly with the understanding that all concerns must be addressed in the revised manuscript and/or in a final detailed point-by-point response (as indicated in your revision plan/ p-b-p-response). Acceptance of your manuscript will depend on a positive outcome of another round of review using the same set of referees.

It is EMBO reports policy to allow a single round of major revision only and acceptance of the manuscript will therefore depend on the completeness of your responses included in the next, final version of the manuscript.

- 1) a .docx formatted version of the final manuscript text (including legends for main figures, EV figures and tables), but without the figures included. Figure legends should be compiled at the end of the manuscript text.
- 2) individual production quality figure files as .eps, .tif, .jpg (one file per figure), of main figures (up to 8) and EV figures (up to 5). Please upload these as separate, individual files upon re-submission.

- 3) one final .docx formatted letter INCLUDING the reviewers' reports and your detailed point-by-point responses to their comments. As part of the EMBO Press transparent editorial process, the point-by-point response is part of the Review Process File (RPF), which will be published alongside your paper.

- 4) a complete author checklist, which you can download from our author guidelines (<https://www.embopress.org/page/journal/14693178/authorguide>). Please insert page numbers in the checklist to indicate where the requested information can be found in the manuscript. The completed author checklist will also be part of the RPF.

- 5) that primary datasets produced in this study (e.g. RNA-seq, ChIP-seq, structural and array data) are deposited in an

appropriate public database. If no primary datasets have been deposited, please also state this in a dedicated section (e.g. 'No primary datasets have been generated and deposited'), see below.

The accession numbers and database should be listed in a formal "Data Availability" section (placed after Materials & Methods) that follows the model below. This is now mandatory (like the COI statement). Please note that the Data Availability Section is restricted to new primary data that are part of this study. This section is mandatory. As indicated above, if no primary datasets have been deposited, please state this in this section

Data availability

8) Regarding data quantification and statistics, please make sure that the number "n" for how many independent experiments were performed, their nature (biological versus technical replicates), the bars and error bars (e.g. SEM, SD) and the test used to calculate p-values is indicated in the respective figure legends (also for potential EV figures and all those in the final Appendix). Please also check that all the p-values are explained in the legend, and that these fit to those shown in the figure. Please provide statistical testing where applicable. Please avoid the phrase 'independent experiment', but clearly state if these were biological or technical replicates. Please also indicate (e.g. with n.s.) if testing was performed, but the differences are not significant. In case n=2, please show the data as separate datapoints without error bars and statistics. See also: <http://www.embopress.org/page/journal/14693178/authorguide#statisticalanalysis>

9) Please also note our reference format:

10) We updated our journal's competing interests policy in January 2022 and request authors to consider both actual and perceived competing interests. Please review the policy <https://www.embopress.org/competing-interests> and update your competing interests if necessary. Please name this section 'Disclosure and Competing Interests Statement' and put it after the Acknowledgements section.

11) We now use CRediT to specify the contributions of each author in the journal submission system. CRediT replaces the author contribution section. Please use the free text box to provide more detailed descriptions and do NOT provide an author contributions section in the revised manuscript text file. See also guide to authors:

<https://www.embopress.org/page/journal/14693178/authorguide#authorshippinguidelines>

12) Please add scale bars of similar style and thickness to microscopic images, using clearly visible black or white bars (depending on the background). Please place these in the lower right corner of the images themselves. Please do not write on or near the bars in the image but define the size in the respective figure legend.

13) Please make sure that all the funding information is also entered into the online submission system and that it is complete

and similar to the one in the acknowledgement section of the manuscript text file.

14) All Materials and Methods need to be described in the main text using our 'Structured Methods' format, which is required for all research articles. According to this format, the Materials and Methods section should include a Reagents and Tools Table (listing key reagents, experimental models, software and relevant equipment and including their sources and relevant identifiers), uploaded as separate file, followed by a Methods and Protocols section in which we encourage the authors to describe their methods using a step-by-step protocol format with bullet points, to facilitate the adoption of the methodologies across labs. More information on how to adhere to this format as well as downloadable templates (.doc or .xls) for the Reagents and Tools Table can be found in our author guidelines (section 'Structured Methods'):

15) Please add up to five keywords to the manuscript and order the manuscript sections like this, using these names:
Title page - Abstract - Keywords - Introduction - Results - Discussion - Methods - Data availability section - Acknowledgements (including funding information) - Disclosure and Competing Interests Statement - References - Figure legends - Expanded View
Figure legends

I look forward to seeing a revised version of your manuscript when it is ready. Please let me know if you have questions or comments regarding the revision.

Best,

Achim Breiling
Senior editor
EMBO reports

Referee #1:

The paper describes a set of experiments to analyse LRRK2 activity in tissues and despite it has very important findings and technical developments is largely descriptive. It does look like a collection of experiments more than a defined hypothesis and experiments to address that.

The flow cytometry assay of the first part is a great technical challenge and represents the establishment of a potentially very useful tool for the field. It would have been important to test other organs, either as controls or for example because of their relevance e.g. lungs. This first part is disconnected from the second part below.

The authors generated a new mouse KI mouse expressing EGFP-LRRK2 and show data the levels of LRRK2 expression are reduced in tissues at different degrees and established a flow cytometry assay to measure LRRK2 expression by monitoring the GFP signal. Interestingly they found that expression does not correlate with activity (as measured by phospho-Rab). I suggest taking this part out as it breaks the flow of the paper. If data using this mouse is included, then microscopy should be included to complement the flow cytometry data. I understand the mice were used later with the anti-CD3 treatment, but it is very confusing that some experiments are done with EGFP-LRRK2 mice and others not. It does look in general like the mice do not behave as wild types and this is an important caveat. Without microscopy of the tissues or even cells (Figure 4) is hard to conclude much about these experiments.

Then the authors show that LRRK2 expression and activity is different in different cell types and depends on inflammation. The anti-CD3 strategy to induce inflammation is very different from physiological inflammation such as sepsis and LPS stimulation, so experiments with other stimuli could be important here to contribute to the message of inflammatory trigger of LRRK2 activation and decoupling of cell type.

The IL-4 data is intriguing but too preliminary. The lack of strong effect of IFN-gamma is expected as the promoter of LRRK2 in mice and humans is different and human cells responds much better with regards to LRRK2 expression after IFN-gamma stimulation.

****Significance:****

The paper describes a set of experiments to analyse LRRK2 activity in tissues and despite it has very important findings and technical developments is largely descriptive. It does look like a collection of experiments more than a defined hypothesis and experiments to address that.

Referee #2:

The paper by Dikovskaya and collaborators investigated the activity and expression of LRRK2 in different subtypes of splenic and intestinal immune cells, taking advantage of a novel GFP-Lrrk2 knockin mouse. Interestingly, they found that T-cell-released IL-4 stimulates Lrrk2 expression in B cells.

I have a few comments and suggestions for the authors.

1. Figure 1C. LRRK2 KO cells display residual Rab10 phosphorylation. Do the authors have any idea of which kinase other than LRRK2 could be involved in this phosphorylation?
2. Since there are no good antibodies for IF/IHC as pointed by the authors, the GFP-Lrrk2 mouse gives the opportunity to check endogenous LRRK2 localization, i.e. in cells untreated or treated with IL-4 or other cytokines. Also, does endogenous GFP-LRRK2 accumulate into filaments/puncta upon MLI2 inhibition? The relocalization into filaments of inhibited LRRK2 has been observed in overexpression but not under endogenous expression. This analysis would be interesting also in light of the observed side effect of type-I inhibitors.
3. Figure 5. The authors need to label more clearly the graphs referring to wt mice versus GFP-Lrrk2 KI mice. They should also replace GFP-LRRK2 with GFP-Lrrk2 since they edited the endogenous murine gene.
4. In the material and methods MLI-2 administration in mice is indicated at 60 mg/kg for 2 hr whereas in suppl. figure 5 the indicated dose is 30 mg/kg. Please correct with the actual dose used.
5. The discovery of IL-4 as a Lrrk2 activator in B cells is a very interesting and novel finding. The authors could take advantage of the GFP tag to investigate LRRK2 interactome upon IL-4 stimulation (optional). Also, is the signaling downstream of IL-4 attenuated in Lrrk2 KO cells?

****Significance:****

The manuscript is well designed and organized, and the experimental approaches are robust. These results are significant for the field as they add additional layers in the complex regulation and regulatory roles of LRRK2 in immunity, with implication for inflammatory disorders and Parkinson's disease.

Referee #3:

The authors present a flow cytometry methodology to assess LRRK2 expression and pathway markers in mouse models and explore LRRK2 in splenic and intestinal immune cells. This is a highly valuable study given the emerging understanding that LRRK2 pathway activity in peripheral tissues may be of crucial importance to Parkinson's disease and Crohn's disease.

P8 : the authors state that their results indicate 'that the effects of LRRK2-R1441C mutation and inflammation on LRRK2 activity represent two different parallel pathways'. This seems like an overinterpretation as pathway suggests the presence of additional partners in the pathway while R1441C is a LRRK2 intrinsic modification. The results can equally be explained by synergistic effects between both activation mechanisms (mutant and inflammation).

Methods and experiment descriptions in results : the authors appear to use the terms anti-CD3 stimulation and CD3 stimulation interchangeably, although it is not always clear in the text that these are synonymous. This should be clarified.

One major observation in this paper is that LRRK2 is not detected in gut epithelial cells as previously has been reported. It would be useful to comment on any differences between the presented protocol and the previous reports, in particular relating to the antigen retrieval step. In order to reinforce the finding, it would be useful to include in situ hybridization data that could further strengthen the observations of which cellular subtypes express LRRK2 and which do not. Indeed, while the KO control shows that there is an unacceptable high non-specific staining, it does not prove absence of expression. Also, can any conclusions be made about expression of LRRK2 in neural cells of the gut? This important information on LRRK2 detection in gut should be mentioned in the abstract and highlighted in the discussion.

The authors mention in the discussion that they 'show for the first time that eosinophils also express active LRRK2 at levels comparable to B-cells and DCs.' The relevance of this finding should be further developed. Why is this important ?

In the isolation of lamina propria cells, what efforts were made to characterize the degree of purification of the lamina propria cells compared to cells of other gut wall layers such as epithelium, muscularis mucosa, or deeper layers? Please specify.

****Minor comments****

Figure 5G, for the graphs indicating LRRK2 activity and LRRK2 phosphorylation, the specific measures should be specified in the graph titles to avoid any ambiguity (pT73-Rab10, pS935-LRRK2).

Suppl figure 1 : please specify the figure label and abbreviation AF568 in the legend.

Suppl figure 2 : please specify the figure label and abbreviation anti-rb in the legend

****Significance:****

The authors present a flow cytometry methodology to assess LRRK2 expression and pathway markers in mouse models and explore LRRK2 in splenic and intestinal immune cells. This is a highly valuable study given the emerging understanding that LRRK2 pathway activity in peripheral tissues may be of crucial importance to Parkinson's disease and Crohn's disease.

We thank the reviewers for going through our manuscript and providing valuable feedback. We are grateful to all 3 reviewers for describing our findings as important and valuable, well-designed and robust, and of value to the Parkinson's and Crohn's disease communities studying LRRK2. Below we detail a point-by-point response to the reviewers' critiques.

Reviewer #1 (Evidence, reproducibility and clarity (Required)):

The paper by Dikovskaya and collaborators investigated the activity and expression of LRRK2 in different subtypes of splenic and intestinal immune cells, taking advantage of a novel GFP-Lrrk2 knockin mouse. Interestingly, they found that T-cell-released IL-4 stimulates Lrrk2 expression in B cells.

I have a few comments and suggestions for the authors.

1) Figure 1C. LRRK2 KO cells display residual Rab10 phosphorylation. Do the authors have any idea of which kinase other than LRRK2 could be involved in this phosphorylation?

As far as we are aware no other kinase is known to phosphorylate Rab10 at T73 *in vivo*. *In vitro*, recombinant Rab10 can be phosphorylated by MST3 at this site (Knebel A. et al, protocols.io <https://dx.doi.org/10.17504/protocols.io.bvjxn4pn>), but its relevance *in vivo* or in cells has not been shown. It is possible that the residual band recognised by anti-pT73 Rab10 ab in splenocytes is unspecific background, as it is mainly seen in LRRK2 KO spleen cells and not in other tissues. But to be certain that our assay assesses LRRK2-dependent Rab10 phosphorylation, we have always compared with the control of LRRK2 inhibition with MLI-2.

2) Since there are no good antibodies for IF/IHC as pointed by the authors, the GFP-Lrrk2 mouse gives the opportunity to check endogenous LRRK2 localization, i.e. in cells untreated or treated with IL-4 or other cytokines. Also, does endogenous GFP-LRRK2 accumulate into filaments/puncta upon MLI2 inhibition? The relocalization into filaments of inhibited LRRK2 has been observed in overexpression but not under endogenous expression. This analysis would be interesting also in light of the observed side effect of type-I inhibitors.

We thank the reviewer for this suggestion. We have now provided live cell images of showing endogenous EGFP-LRRK2 expression in B cells (Fig. 5H). However, the signal is very weak, as based on our proteomics data, even IL-4 stimulated B cells only express around 7000 copies of LRRK2 per cell (Fig. 5E). No obvious localisation was visible in the images, although it does appear that in IL-4 treatment, more of LRRK2 is diffuse cytoplasmic. We also tried to optimise staining of EGFP-LRRK2 with anti-GFP antibodies. Treatment with MLI-2 did not change the staining pattern in B cells. We include these data here, however since no change was seen, we did not include these images in the main text.

Figure R1. Localization of endogenous EGFP-LRRK2 In B cells upon treatment with IL-4 for 24h and LRRK2 inhibitor MLI-2 for the last 2 hours. Splenic B cells from WT or EGFP-Lrrk2-KI mice were treated with IL-4 +/- MLI-2 and then purified, fixed, permeabilized and stained with AlexaFluor647 labelled anti-GFP booster (white) and DAPI (blue). Cells were mounted onto a coverslip and then visualized by confocal microscopy. Scalebar Lengths: 4 μ m

3) Figure 5. The authors need to label more clearly the graphs referring to wt mice versus GFP-Lrrk2 KI mice.

As suggested, we have now labelled the panels referring to the WT mice only with “WT mice”, to distinguish them from the other panels that incorporate data from both EGFP-Lrrk2 mice and their WT littermates used as a background.

They should also replace GFP-LRRK2 with GFP-Lrrk2 since they edited the endogenous murine gene.

Thank you, we have corrected the EGFP-Lrrk2, and the other mouse genotypes.

4) In the material and methods MLI-2 administration in mice is indicated at 60 mg/kg for 2 hr whereas in suppl. figure 5 the indicated dose is 30 mg/kg. Please correct with the actual dose used.

Thank you, we have corrected the mistake.

5) The discovery of IL-4 as a Lrrk2 activator in B cells is a very interesting and novel finding. The authors could take advantage of the GFP tag to investigate LRRK2 interactome upon IL-4 stimulation (optional). Also, is the signaling downstream of IL-4 attenuated in Lrrk2 KO cells?

We thank the reviewer for these interesting suggestions. The role of LRRK2 in IL-4 activated B-cells is currently under active research in the lab.

Reviewer #2 (Significance (Required)):

The manuscript is well designed and organized, and the experimental approaches are robust. These results are significant for the field as they add additional layers in the complex regulation and regulatory roles of LRRK2 in immunity, with implication for inflammatory disorders and Parkinson's disease.

We thank the reviewer for their positive comments and for recognising our efforts to provide some clarity to a complex field.

Reviewer #2 (Evidence, reproducibility and clarity (Required)):

The authors present a flow cytometry methodology to assess LRRK2 expression and pathway markers in mouse models and explore LRRK2 in splenic and intestinal immune cells. This is a highly valuable study given the emerging understanding that LRRK2 pathway activity in peripheral tissues may be of crucial importance to Parkinson's disease and Crohn's disease.

P8 : the authors state that their results indicate 'that the effects of LRRK2-R1441C mutation and inflammation on LRRK2 activity represent two different parallel pathways'. This seems like an overinterpretation as pathway suggests the presence of additional partners in the pathway while R1441C is a LRRK2 intrinsic modification. The results can equally be explained by synergistic effects between both activation mechanisms (mutant and inflammation).

We agree with the reviewer, and have added this into the text. The sentence now reads “suggesting that the LRRK2-R1441C mutation and inflammation have different impacts on LRRK2 activity, either in parallel or in synergy.”

Methods and experiment descriptions in results : the authors appear to use the terms anti-CD3 stimulation and CD3 stimulation interchangeably, although it is not always clear in the text that these are synonymous. This should be clarified.

We thank reviewer for pointing out this error on our part. We have made the necessary changes to always refer to the stimulation as anti-CD3.

One major observation in this paper is that LRRK2 is not detected in gut epithelial cells as previously has been reported. It would be useful to comment on any differences between the presented protocol and the previous reports, in particular relating to the antigen retrieval step. In order to reinforce the finding, it would be useful to include in situ hybridization data that could further strengthen the observations of which cellular subtypes express LRRK2 and which do not. Indeed, while the KO control shows that there is an unacceptable high non-specific staining, it does not prove absence of expression. Also, can any conclusions be made about expression of LRRK2 in neural cells of the gut? This important information on LRRK2 detection in gut should be mentioned in the abstract and highlighted in the discussion.

We thank the reviewer for pointing this out. In fact, we think the observation that LRRK2 is not detected in epithelial cells is so important that we have a separate publication exploring this point. We have now added a citation to this publication (Tasegian *et al*, 2024) in the text. In this manuscript we have explored the expression of LRRK2 in human and murine intestinal epithelial cells using qPCR. Although we do not have in situ hybridization data, we believe that using both the EGFP-LRRK2 and the pRab10 flow cytometry, as well as qPCR and proteomics on selected cell types, corroborates our findings on the cell types that express LRRK2. Interestingly, during the process of our revisions, another study was published (Sun *et al*, 2024) that performed in situ hybridization in murine and human guts, and also showed no binding in epithelial cells, but in some cells in the lamina propria, assumed to be myeloid cells. We did not analyse LRRK2 expression in the neural cells of the gut, as our focus was on the immune cells, however we hope that others will use the tools developed here to explore this further.

The authors mention in the discussion that they 'show for the first time that eosinophils also express active LRRK2 at levels comparable to B-cells and DCs.' The relevance of this finding should be further developed. Why is this important?

We thank the reviewer for this point. We don't know how LRRK2 is important in these cells. However, as the role of LRRK2 in eosinophils and neutrophils has not yet been explored and both cell types play important roles in IBD, we think it is important to point out. We have now added a sentence to the discussion highlighting the importance of eosinophils in IBD. "Since eosinophils have recently been implicated as key player in intestinal defense and colitis (Gurtner *et al*, 2022), it will be interesting to evaluate LRRK2 functions in these cells."

In the isolation of lamina propria cells, what efforts were made to characterize the degree of purification of the lamina propria cells compared to cells of other gut wall layers such as epithelium, muscularis mucosa, or deeper layers? Please specify.

Isolation of lamina propria cells is a very well-established process (LeFrancois and Lycke, 'Isolation of Mouse Small Intestinal Intraepithelial Lymphocytes, Peyer's Patch, and Lamina Propria Cells.' *Curr. Protocols in Immunology* 2001), where we extensively wash off the epithelial layer before digesting the tissue for the LP. We have checked in the past that we do not get epithelial contamination in lamina propria preparations by staining for CD103 which is only expressed on intraepithelial cells. After the enzymatic digestion of the remaining tissue, the muscle and wall of the gut are still intact, so we do not get any contamination with other deeper layers. The subsets of cells we find in the LP are in line with isolations from other labs.

Minor comments

Figure 5G, for the graphs indicating LRRK2 activity and LRRK2 phosphorylation, the specific measures should be specified in the graph titles to avoid any ambiguity (pT73-Rab10, pS935-LRRK2).

We have added the specifications to the new version of the figure.

Suppl figure 1 : please specify the figure label and abbreviation AF568 in the legend.
Suppl figure 2 : please specify the figure label and abbreviation anti-rb in the legend

Thank you, we added the abbreviations to the legends of Figure EV1 that now combines these figures. The Figure labels for both figures have been already included at the top of figure legends.

Reviewer #1 (Significance (Required)):

The authors present a flow cytometry methodology to assess LRRK2 expression and pathway markers in mouse models and explore LRRK2 in splenic and intestinal immune cells. This is a highly valuable study given the emerging understanding that LRRK2 pathway activity in peripheral tissues may be of crucial importance to Parkinson's disease and Crohn's disease.

We thank the reviewer for recognising the value of this study.

Reviewer #1

Evidence, reproducibility and clarity

The paper describes a set of experiments to analyse LRRK2 activity in tissues and despite it has very important findings and technical developments is largely descriptive. It does look like a collection of experiments more than a defined hypothesis and experiments to address that.

We thank the reviewer for recognising the importance of our findings and the technical developments. We agree that the paper's focus is to describe where LRRK2 is expressed in immune cells, and in which cells is it active or activated after inflammation in a hypothesis-free unbiased manner. We believe this is important data to share as a resource for the wider LRRK2 community and we will submit the manuscript as a Resource.

The flow cytometry assay of the first part is a great technical challenge and represents the establishment of a potentially very useful tool for the field. It would have been important to test other organs, either as controls or for example because of their relevance e.g. lungs. This first part is disconnected from the second part below.

We thank the reviewer for pointing out that the pRab10 assay would be useful to apply to other organs too. Since we are interested in the role of LRRK2 in IBD, we had focused on applying the pRab10 assay on intestinal tissue, with spleens also analysed as major lymphoid organ and a source of immune cells that can translocate to the gut in inflammation. We hope that the publication of this method would allow other researchers to analyse other tissues in the future.

The authors generated a new mouse KI mouse expressing EGFP-LRRK2 and show data the levels of LRRK2 expression are reduced in tissues at different degrees and established a flow cytometry assay to measure LRRK2 expression by monitoring the GFP signal. Interestingly they found that expression does not correlate with activity (as measured by phospho-Rabs). I suggest taking this part out as it breaks the flow of the paper. If data using this mouse is included, then microscopy should be included to complement the flow cytometry data. I understand the mice were used later with the anti-CD3 treatment, but it is very confusing that some experiments are done with EGFP-LRRK2 mice and others not. It does look in general like the mice do not behave as wild types and this is an important caveat. Without microscopy of the tissues or even cells (Figure 4) is hard to conclude much about these experiments.

We thank the reviewer for this point and would like to explain. It is true that in Suppl Figure 5 (now Figure EV2), we show reduction of LRRK2 signal in the EGFP-Lrrk2-KI mice. However, based on immunoblotting, a significant reduction in EGFP-LRRK2 expression levels was seen only in the brain, but not in the tissues we analysed, that is the spleen and the intestine. Further, we have shown clearly using proteomics (Fig. 3D and 5E), that the GFP signal in immune cells correlates

very well with the WT LRRK2 expression. Therefore, we think that the GFP signal in these mice reflects WT LRRK2 expression pattern.

Despite the limitations of reduced kinase activity of EGFP-LRRK2 that we thoroughly describe, we think this model is very useful since no antibodies work to stain for LRRK2 in mice. We therefore respectfully disagree with this reviewer that the EGFP-LRRK2 data should be taken out, as it has proven to be an invaluable tool to measure and track changes in endogenous LRRK2 expression. Moreover, we think the fact that LRRK2 expression does not correlate with levels of activity, that is, LRRK2 is more active in some immune cells than in others, is a very important finding that evidences the cell-specific regulation of LRRK2 activity beyond its expression level.

We tried but were unable to visualize the EGFP-LRRK2 signal using fluorescence microscopy in the tissue. This is most likely due to the low expression of LRRK2 as proteomics data suggests that even neutrophils express less than 9000 copies of LRRK2 (www.immpres.co.uk, data from (Sollberger *et al*, 2024)). Detection of the EGFP signal is confounded further by the high background autofluorescence in this channel in tissues, particularly in the gut. We now explain the lack of tissue images from the EGFP-LRRK2 mice in the text. However, we have visualized the EGFP-LRRK2 in B cells (Fig. 5H), where the signal has visibly increased post IL-4 treatment. The signal was still very weak, and we did not observe any changes in localisation of LRRK2 in the cells.

We have also added the following paragraph to the discussion:

“We complemented the pRab10 assay with the development of the EGFP-Lrrk2-KI reporter mouse. Although the reporter was initially designed as a fluorescent tracker for imaging LRRK2 localisation in cells and tissues, the low expression of LRRK2, combined with high and variable autofluorescence in tissues precluded its use for microscopy. Even in neutrophils, which express highest level of LRRK2 among immune cells, there are less than 9000 copies of LRRK2 per cell (Sollberger *et al*, 2024), making it difficult to identify localization. However, the EGFP signal was sufficient for flow cytometry-based measurements, where background autofluorescence of each cell type was taken into account and subtracted.”

Then the authors show that LRRK2 expression and activity is different in different cell types and depends on inflammation. The anti-CD3 strategy to induce inflammation is very different from physiological inflammation such as sepsis and LPS stimulation, so experiments with other stimuli could be important here to contribute to the message of inflammatory trigger of LRRK2 activation and decoupling of cell type.

We thank the reviewer for this suggestion. We used the anti-CD3 model as it also causes intestinal inflammation, and mimics T-cell cytokine storms that happens in many diseases, but we see the value of measuring other forms of inflammation. We have now added data for LRRK2-induced pRab10 in zymosan-induced peritonitis (Figure EV3 D-E). Here we show that pRab10 increase in B cells and in neutrophils in the spleen and peritoneal lavage. Surprisingly we see that pRab10 is decreased in the macrophages that migrate to the peritoneal cavity after peritonitis. This interesting finding indicates that there is much to be learnt about how LRRK2 is regulated in different subsets of immune cells.

The IL-4 data is intriguing but too preliminary. The lack of strong effect of IFN-gamma is expected as the promoter of LRRK2 in mice and humans is different and human cells responds much better with regards to LRRK2 expression after IFN-gamma stimulation.

We are confused by what the reviewer means by saying the IL-4 data is preliminary. We have shown by flow cytometry, immunoblotting, qPCR and proteomics that IL-4 induced LRRK2 expression in B-cells. So we are uncertain as to how else this can be shown. As to the effect of IFN γ on LRRK2 expression, it may indeed be that human cells respond better than murine cells. Importantly, the IL-4 ability to induce LRRK2 in B-cells is a novel and important finding, regardless of the effects of IFN γ .

Dear Dr. Swamy,

Thank you for the submission of your revised manuscript to our editorial offices. I have now received the report from the three referees that were asked to re-evaluate the study, you will find below. As you will see, the referees now fully support the publication of the study in EMBO reports.

Before I can proceed with formal acceptance, I have these editorial requests I ask you to address in a final revised manuscript:

- Please name the Methods section just 'Methods'.

- EV Figs. EV1 and EV2 have more than one page. We can't process figures that have more than one page. Please rearrange this to have figures (main and EV figures) with one page only. See also our guide for figure preparation: http://wol-prod-cdn.literatumonline.com/pb-assets/embo-site/EMBOPress_Figure_Guidelines_061115-1561436025777.pdf

We could accommodate one or two more EV figures. Otherwise, remaining data needs to be moved to the Appendix.

- The legend for Fig. EV1 contains 2 different descriptions for panel B, but no description for panel A. Please check.

- Please make sure that all the funding information is also entered into the online submission system and that it is complete and similar to the one in the acknowledgement section of the manuscript text file. Presently grant SCAF/18/01 is missing in the manuscript text file (Acknowledgements) and grants from the pharmaceutical companies supporting the Division of Signal Transduction Therapy Unit (Boehringer Ingelheim, GlaxoSmithKline, and Merck KGaA) and GlaxoSmithKline are missing in the submission system. Please check.

- Please check again that the number "n" for how many independent experiments were performed, their nature (biological versus technical replicates), the bars and error bars (e.g. SEM, SD) and the test used to calculate p-values is indicated in the respective figure legends. Please also check that all the p-values are explained in the legend, and that these fit to those shown in the figure. Please provide statistical testing where applicable. Please avoid the phrase 'independent experiment', but clearly state if these were biological or technical replicates. Please also indicate (e.g. with n.s.) if testing was performed, but the differences are not significant. In case n=2, please show the data as separate datapoints without error bars and statistics. See also:

<http://www.embopress.org/page/journal/14693178/authorguide#statisticalanalysis>

If n<5, please show single datapoints for diagrams. Presently, several diagrams lack statistic, show only partial statistics or the 'n.s.' is missing. Please check. Moreover:

- Please note that the exact p values are not provided in the legends of figures 2F, 3A-D; 4A, B, C, D, F, G; 5 B, F; EV3 A, E.

- Please indicate the statistical test used for data analysis in the legends of figures 1H, I; 5B, C, E; EV2 C, EV3 E.

- Please indicate what */ **/ ***/ **** represents; if this represents p value(s), please specify the exact p value in the legend(s) of figure(s) EV3 F, G.

- Please indicate what */ **/ ***/ **** represents; if this represents p value(s), please indicate the statistical test used and where appropriate, specify the exact p value in the legend(s) of figure(s) 5D, EV2 A, B.

- Please note that information related to n is missing in the legends of figures 1G, 2D, E, F; 3E, 4C, F, G; 5B, E, F; EV2 A-D; EV3 F, G, H; EV4 A, B.

- Please note that the error bars are not defined in the legends of figures EV2 D, EV3D, E.

- Please add to each legend (main, EV and Appendix figures, where applicable) a 'Data Information' section explaining the statistics used or providing information regarding replicates and scales. See:

- Please add scale bars of similar style and thickness to microscopic images, using clearly visible black or white bars (depending on the background). Please place these in the lower right corner of the images themselves. Please do not write on or near the bars in the image but define the size in the respective figure legend. Presently, some scale bars have text nearby.

- Please remove the template text and the example table from the author checklist.

- Please move the antibody and primer tables from the Methods section to the Reagents and Tools table. Please also add callouts for the table to the Methods section where applicable.

- Please remove now the referee access information from the Data Availability section and make sure the datasets are public latest upon online publication of the paper.

- During our routine figure check, we noted a duplicated area in 2 images of panel A of Fig. EV1 (3rd and 4th image, second row - see the attached figure check report). If this is intentional, please indicate and explain this reuse in the respective legend. Moreover, the images in the column 'DAPI anti-mouse' of panel EV1D contains writing underneath the figures (see again the attached figure check report). Could you please explain this and provide images with the text fragments.

In addition, I would need from you uploaded separately:

Best,

Referee #1:

Thanks to the authors for addressing my concerns and comments. I still have reservations that the authors can only see the EGFP signal from EGFP-LRRK2 by flow cytometry and not imaging- it seems that having the mice to use only for flow cytometry will not be very useful. However, I see the efforts they put in the revision and I recommend publication in EMBO Reports.

Referee #2:

The authors successfully addressed the comments and the manuscript is now suitable for publication

Referee #3:

All of my concerns have been adequately addressed. I look forward to the publication.

All editorial and formatting issues were resolved by the authors.

Dr. Mahima Swamy
University of Dundee
MRC Protein Phosphorylation and Ubiquitylation Unit
Dow Street
Dundee, Angus DD1 5EH
United Kingdom

Dear Dr. Swamy,

I am very pleased to accept your manuscript for publication in the next available issue of EMBO reports. Thank you for your contribution to our journal.

Yours sincerely,
